# Unknown-Aware Domain Adversarial Learning for Open-Set Domain Adaptation

**JoonHo Jang**
KAIST
adkto8093@kaist.ac.kr

**Byeonghu Na**
KAIST
wp03052@kaist.ac.kr

**DongHyeok Shin**
KAIST
tlsehdgur0@kaist.ac.kr

**Mingi Ji** *
KAIST
qwertgfdcvb@kaist.ac.kr

**Kyungwoo Song**
University of Seoul
kyungwoo.song@uos.ac.kr

**Il-Chul Moon**
KAIST, Summary.AI
icmoon@kaist.ac.kr

## Abstract

Open-Set Domain Adaptation (OSDA) assumes that a target domain contains unknown classes, which are not discovered in a source domain. Existing domain adversarial learning methods are not suitable for OSDA because distribution matching with *unknown* classes leads to negative transfer. Previous OSDA methods have focused on matching the source and the target distribution by only utilizing *known* classes. However, this *known*-only matching may fail to learn the target-*unknown* feature space. Therefore, we propose Unknown-Aware Domain Adversarial Learning (UADAL), which *aligns* the source and the target-*known* distribution while simultaneously *segregating* the target-*unknown* distribution in the feature alignment procedure. We provide theoretical analyses on the optimized state of the proposed *unknown-aware* feature alignment, so we can guarantee both *alignment* and *segregation* theoretically. Empirically, we evaluate UADAL on the benchmark datasets, which shows that UADAL outperforms other methods with better feature alignments by reporting state-of-the-art performances[†].

## 1 Introduction

*Unsupervised Domain Adaptation* (UDA) means leveraging knowledge from a labeled source domain to an unlabeled target domain [4, 3, 2, 14, 8]. This adaptation implicitly assumes the source and the target data distributions, where each distribution is likely to be drawn from different distributions, i.e., *domain shift* (see Figure 1a). Researchers have approached the modeling of two distributions by statistical matching [20, 27, 15, 16], or domain adversarial learning [7, 28, 18]. Among them, *domain adversarial learning* has been successful in matching between the source and the target distributions via feature alignment, so the model can accomplish the domain-invariant representations.

There is another dynamic aspect of the source and the target distribution. In a realistic scenario, these distributions may expand a class set, which is called *unknown* classes. This expansion creates a field of Open-Set Domain Adaptation (OSDA) [23, 13]. Existing adversarial learning methods of UDA have limitations to solve OSDA because matching the source and the target distribution with the *unknown* classes may lead to the negative transfer [6] due to the class set mismatch (see Figure 1b).

Previous OSDA methods focused on matching between the source and the target domain only embedded in the *known* class set via domain adversarial learning [23, 13]. However, this *known-only* feature alignment may fail to learn the target-unknown feature space because of no alignment

---

*now at Google (mingiji@google.com)

[†]The code will be publicly available on `https://github.com/JoonHo-Jang/UADAL`.

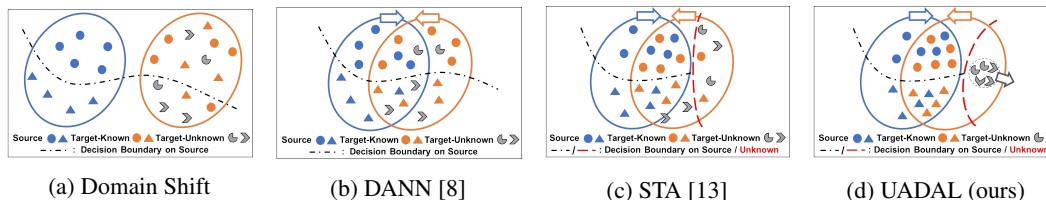

Figure 1: The feature distributions from the source and the target domain, with the decision boundaries. The blue/orange/gray arrows represent the alignment signal on the source/target-known/target-unknown domain.

signal from the target-unknown instances. Therefore, a classifier is not able to learn a clear decision boundary for *unknown* classes because the target-unknown instances are not segregated enough in the aligned feature space (see Figure 1c). On the other hand, some OSDA methods propose to learn intrinsic target structures by utilizing self-supervised learning without distribution matching [24, 12]. However, this weakens the model performance under large domain shifts. Therefore, in order to robustly solve OSDA, distribution matching via the domain adversarial learning is required, and the class set mismatch should be resolved to prevent the negative transfer, simultaneously.

This paper proposes a new domain adversarial learning for OSDA, called Unknown-Aware Domain Adversarial Learning (UADAL). Specifically, we aim at enforcing the target-unknown features to *move apart* from the source and the target-known features, while aligning the source and the target-known features. We call these alignments as the *segregation* of the target-unknown features (gray-colored arrow in Figure 1d). Although this distribution *segregation* is an essential part of OSDA, the segregation has been only implicitly modeled because of its identification on the target-unknown instances. UADAL is the first explicit mechanism to simultaneously align and segregate the three sets (source, target-known, and target-unknown instances) via the domain adversarial learning. Therefore, the proposed *unknown-aware* feature alignment enables a classifier to learn a clear decision boundary on both *known* and *unknown* class in the feature space.

UADAL consists of three novel mechanisms. First, we propose a new domain discrimination loss to engage the target-unknown information. Second, we formulate a sequential optimization to enable the unknown-aware feature alignment, which is suitable for OSDA. We demonstrate that the optimized state of the proposed alignment is theoretically guaranteed. Third, we also suggest a posterior inference to effectively recognize target-*known/unknown* information without any thresholding.

## 2 Preliminary

### 2.1 Open-Set Domain Adaptation

This section provides a formal definition of OSDA. The fundamental properties are two folds: 1) the different data distributions of a source domain, $p_s(x, y)$, and a target domain, $p_t(x, y)$; and 2) the additional classes of the target domain, which were not observed in the source domain. Specifically, we define the source and the target datasets as $\chi_s = \{(x_s^i, y_s^i)\}_{i=1}^{n_s}$ and $\chi_t = \{(x_t^i, y_t^i)\}_{i=1}^{n_t}$, respectively. $y_t^i$ is not available at training under the UDA setting. Additionally, we designate $\mathcal{C}_s$ to be the source class set (a.k.a. a shared-*known* class), and $\mathcal{C}_t$ to be the target class set, i.e., $y_t^i \in \mathcal{C}_t$. OSDA dictates $\mathcal{C}_s \subset \mathcal{C}_t$ [23]. $\mathcal{C}_t \setminus \mathcal{C}_s$ is called *unknown* classes. In spite that there can be multiple *unknown* classes, we consolidate $\mathcal{C}_t \setminus \mathcal{C}_s$ as $y_{unk}$ to be a single *unknown* class, due to no knowledge on $\mathcal{C}_t \setminus \mathcal{C}_s$.

The learning objectives for OSDA become both 1) the optimal class classification in $\chi_t$ if a target instance belongs to $\mathcal{C}_s$, and 2) the optimal *unknown* classification if a target instance belongs to $\mathcal{C}_t \setminus \mathcal{C}_s$. This objective is formulated as follows, with a classifier $f$ and the cross-entropy loss function $\mathcal{L}_{ce}$,

$$\min_f \mathbb{E}_{p_t(x,y)}[\mathbf{1}_{y_t \in \mathcal{C}_s} \mathcal{L}_{ce}(f(x_t), y_t) + \mathbf{1}_{y_t \in \{\mathcal{C}_t \setminus \mathcal{C}_s\}} \mathcal{L}_{ce}(f(x_t), y_{unk})]. \tag{1}$$

### 2.2 Adversarial Domain Adaptation

We first step back from OSDA to Closed-set DA (CDA) with $\mathcal{C}_t \setminus \mathcal{C}_s = \phi$, where the domain adversarial learning is widely used. Domain-Adversarial Neural Network (DANN) [7] proposes,

$$\min_{f,G} \max_D \{\mathbb{E}_{p_s(x,y)}[\mathcal{L}_{ce}(f(G(x_s)), y_s)] - \mathcal{L}_d\}. \tag{2}$$

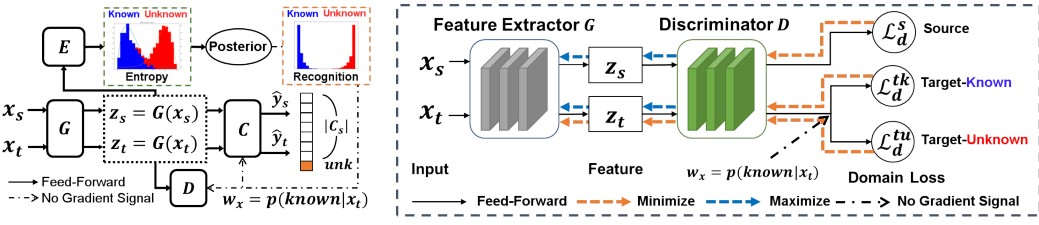

(a) The overall structure of UADAL       (b) The optimization procedure of UADAL

Figure 2: Overview of the proposed Unknown-Aware Domain Adversarial Learning (UADAL) approach.

This objective assumes $G(x)$ to be a feature extractor that learns the domain-invariant features, enabled by the minimax game with a domain discriminator $D$, with respect to $\mathcal{L}_d$, as below.

$$\mathcal{L}_d = \mathbb{E}_{p_s(x)}\left[-\log D_s(G(x))\right] + \mathbb{E}_{p_t(x)}[-\log D_t(G(x))] \tag{3}$$

The adversarial framework adapts $G(x)$ toward the indistinguishable feature distributions between the source and the target domain by the minimax game on $D(G(x)) = [D_s(G(x)), D_t(G(x))]$. Here, $D$ is set up to have a two-dimensional output to indicate either source or target domain, denoted as $D_s$ and $D_t$, respectively. Given these binarized outputs of $D$, this formulation is not appropriate to differentiate the target-known and the target-unknown features in OSDA. DANN enforces to include the target-*unknown* features in the distribution matching, which leads to performance degradation by negative transfer [6]. Figure 1b represents this undesired alignment of *unknown* classes.

**Separate To Adapt (STA)** [13] proposes a weighted domain adversarial learning, to resolve the negative transfer by utilizing a weighting scheme. Therefore, STA modifies $\mathcal{L}_d$ as follows,

$$\mathcal{L}_d^{\text{STA}} = \mathbb{E}_{p_s(x)}\left[-\log D_s(G(x))\right] + \mathbb{E}_{p_t(x)}[-w_x \log D_t(G(x))], \tag{4}$$

where $w_x$ represents a probability of an instance $x$ belonging to the shared-known classes. $\mathcal{L}_d^{\text{STA}}$ enables the domain adversarial learning to align the features from the source and the most likely target-known instances with the high weights. However, the feature extractor, $G$, is not able to move apart the target-unknown instances due to lower weights, i.e., no alignment signals (see Figure 1c). STA modeled this known-only feature alignment by assuming the unknown feature would be implicitly learned through the estimation on $w_x$. Besides learning $w_x$, the target-unknown instances cannot contribute to the training of STA because their loss contribution will be limited by lower $w_x$, in the second term of Eq. (4). Therefore, STA does not take information from the target-unknowns, and later Section 4.3 empirically shows that the target-unknown segregation of STA is limited.

## 2.3 Comparison to Recent OSDA Research

In terms of the domain adversarial learning, in addition to STA, OSBP [23] utilizes a classifier to predict the target instances to the pre-determined threshold, and trains the feature extractor to deceive the classifier for aligning to *known* classes or rejecting as *unknown* class. However, their recognition on *unknown* class only depends on the threshold value, without considering the data instances. PGL [19] introduces progressive graph-based learning to regularize the class-specific manifold, while jointly optimizing the feature alignment via domain adversarial learning. However, their adversarial loss includes all instances of the target domain, which is critically weakened by the negative transfer.

On the other hand, self-supervised learning approaches have been proposed recently, in order to exploit the intrinsic structure of the target domain [5, 24, 12]. However, these approaches do not have any feature alignments between the source and the target domain, which leads to performance degradation under significant domain shifts. There is also a notable work, OSLPP [31] optimizing projection matrix toward a common subspace to class-wisely align the source and the target domain. However, the optimization requires pair-wise distance calculation, which results in a growing complexity of $O(n^2)$ by the $n$ data instances. It could be limited to the large-scale domain. We provide detailed comparisons and comprehensive literature reviews in Appendix A.

# 3   Methodology

## 3.1   Overview of UADAL

Figure 2a illustrates the neural network compositions and the information flow. Our model consists of four networks: 1) a feature extractor, $G$; 2) a domain discriminator, $D$; 3) an open-set recognizer, $E$; and 4) a class classifier network, $C$. The networks $G$, $D$, $E$, and $C$ are parameterized by $\theta_g, \theta_d, \theta_e$, and $\theta_c$, respectively. First, we propose a new domain discrimination loss, and we formulate the sequential optimization problem, followed by theoretical analyses. In order to recognize target-*unknown* information, we introduce a posterior inference method, followed by open-set classification.

## 3.2   Sequential Optimization Problem for Unknown-Aware Feature Alignment

**Model Structure** We assume that we identify three domain types, which are 1) the source (*s*), 2) the target-known (*tk*), and 3) and target-unknown (*tu*). As the previous works such as STA and DANN utilize a two-way domain discriminator with $(D_s, D_t)$, they can treat only *s* and *tk* (or *t*). However, a domain discriminator in OSDA should be able to handle *tu* information for *segregation*. Therefore, we designed the discriminator dimension to be three $(D_s, D_{tk}, D_{tu})$, as follows,

$$D(G(x)) = [D_s(G(x)), D_{tk}(G(x)), D_{tu}(G(x))]. \tag{5}$$

Given these three dimensions of $D$, we propose new domain discrimination losses, $\mathcal{L}_d^s$ and $\mathcal{L}_d^t$, as

$$\mathcal{L}_d^s(\theta_g, \theta_d) = \mathbb{E}_{p_s(x)} \left[ -\log D_s(G(x)) \right], \tag{6}$$

$$\mathcal{L}_d^t(\theta_g, \theta_d) = \mathbb{E}_{p_t(x)} [ -w_x \log D_{tk}(G(x)) - (1 - w_x) \log D_{tu}(G(x))], \tag{7}$$

where $w_x := p(known|x)$, named as *open-set recognition*, is the probability of a target instance, $x$, belonging to a shared-*known* class. We introduce the estimation of this probability, $w_x$, in the section 3.4 later. From $\mathcal{L}_d^t$ in Eq. (7), our modified $D$ becomes to be able to discriminate *tk* and *tu*, explicitly.

Our new unknown awareness of $D$ allows us to segregate the target-unknown features via the domain adversarial learning framework. Firstly, we decompose $\mathcal{L}_d^t(\theta_g, \theta_d)$ into $\mathcal{L}_d^{tk}(\theta_g, \theta_d)$ and $\mathcal{L}_d^{tu}(\theta_g, \theta_d)$,

$$\mathcal{L}_d^t(\theta_g, \theta_d) = \mathcal{L}_d^{tk}(\theta_g, \theta_d) + \mathcal{L}_d^{tu}(\theta_g, \theta_d), \tag{8}$$

$$\mathcal{L}_d^{tk}(\theta_g, \theta_d) := \lambda_{tk} \mathbb{E}_{p_{tk}(x)} \left[ -\log D_{tk}(G(x)) \right], \tag{9}$$

$$\mathcal{L}_d^{tu}(\theta_g, \theta_d) := \lambda_{tu} \mathbb{E}_{p_{tu}(x)} \left[ -\log D_{tu}(G(x)) \right], \tag{10}$$

where $p_{tk}(x) := p_t(x|known)$, $p_{tu}(x) := p_t(x|unknown)$, $\lambda_{tk} := p(known)$, and $\lambda_{tu} := p(unknown)$. The derivation of Eq. (8) is based on $p_t(x) = \lambda_{tk} p_{tk}(x) + \lambda_{tu} p_{tu}(x)$, which comes from the law of total probability (Details in Appendix B.1.1). Therefore, this decomposition of $\mathcal{L}_d^t$ into $\mathcal{L}_d^{tu}$ and $\mathcal{L}_d^{tk}$ enables the different treatments on *tk* and *tu* feasible. The three-way discriminator and its utilization in Eq. (9)-(10) becomes the unique contribution from UADAL.

**Optimization** Our goal of this unknown-aware domain discrimination is to align the features from the source and the target-known instances while simultaneously *segregating* the target-unknown features. To achieve this goal, we propose a new sequential optimization problem w.r.t. $G$ and $D$. Based on the losses, $\mathcal{L}_d^s(\theta_g, \theta_d)$, $\mathcal{L}_d^{tk}(\theta_g, \theta_d)$, and $\mathcal{L}_d^{tu}(\theta_g, \theta_d)$, we formulate the optimization problem as,

$$\min_{\theta_d} \mathcal{L}_D(\theta_g, \theta_d) = \mathcal{L}_d^s(\theta_g, \theta_d) + \mathcal{L}_d^{tk}(\theta_g, \theta_d) + \mathcal{L}_d^{tu}(\theta_g, \theta_d), \tag{11}$$

$$\max_{\theta_g} \mathcal{L}_G(\theta_g, \theta_d) = \mathcal{L}_d^s(\theta_g, \theta_d) + \mathcal{L}_d^{tk}(\theta_g, \theta_d) - \mathcal{L}_d^{tu}(\theta_g, \theta_d). \tag{12}$$

This alternating objective by Eq. (11) and Eq. (12) is equivalent to the adversarial domain adaptation models [7, 13] to learn the domain-invariant features. Unlike the previous works, however, Eq. (11) represents that we train the domain discriminator, $D$, to classify an instance as either source, target-known, or target-unknown domain. In terms of the feature extractor, $G$, we propose Eq. (12) to maximize the domain discrimination loss for the source and the target-known domain while minimizing the loss for the target-unknown domain. From the different signals on $\mathcal{L}_d^{tk}$ and $\mathcal{L}_d^{tu}$, we only treat the adversarial effects on $\mathcal{L}_d^s$ and $\mathcal{L}_d^{tk}$. Meanwhile, minimizing $\mathcal{L}_d^{tu}$ provides for the network $G$ to learn the discriminative features on *tu*. Eventually, $G$ and $D$ align the source and target-known features and segregate the target-unknown features from the source and the target-known features (the optimization details in Figure 2b). We provide the theoretic analysis of the optimized state of the proposed feature alignment in the next subsection, which is unexplored in the OSDA field.

## 3.3 Theoretic Analysis of Sequential Optimization

This section provides a theoretic discussion on the proposed feature alignment by our sequential optimization, w.r.t. Eq. (11)-(12). Since our ultimate goal is training $G$ to learn unknown-aware feature alignments, we treat $G$ as a *leader* and $D$ as a *follower* in the game theory. We first optimize Eq. (11) to find $D^*$ with a fixed $G$. The optimal $D^*$ given the fixed $G$ results in the below output :

$$D^*(z) = \left[ \frac{p_s(z)}{2p_{avg}(z)}, \frac{\lambda_{tk}p_{tk}(z)}{2p_{avg}(z)}, \frac{\lambda_{tu}p_{tu}(z)}{2p_{avg}(z)} \right], \tag{13}$$

where $p_{avg}(z) = (p_s(z) + \lambda_{tk}p_{tk}(z) + \lambda_{tu}p_{tu}(z))/2$ (Details in Appendix B.1.2). Here, $z \in \mathcal{Z}$ stands for the feature space from $G$, i.e., $p_d(z) = \{G(x;\theta_g)|x \sim p_d(x)\}$ where $d$ is $s$, $tk$, or $tu$. Given $D^*$ with the optimal parameter $\theta_d^*$, we optimize $G$ in Eq. (12). Here, we show that the optimization w.r.t. $G$ is equivalent to the weighted summation of KL Divergences, by Theorem 3.1.

**Theorem 3.1.** *(Proof in Appendix B.1.3) Let $\theta_d^*$ be the optimal parameter of $D$ by optimizing Eq. (11). Then, $-\mathcal{L}_G(\theta_g, \theta_d^*)$ can be expressed as, with a constant $C_0$,*

$$-\mathcal{L}_G(\theta_g, \theta_d^*) = D_{KL}(p_s \| p_{avg}) + \lambda_{tk}D_{KL}(p_{tk} \| p_{avg}) - \lambda_{tu}D_{KL}(p_{tu} \| p_{avg}) + C_0. \tag{14}$$

We note that $p_s, p_{tk}$, and $p_{tu}$ are the feature distribution of each domain, mapped by $G$, respectively. Therefore, we need to minimize Eq. (14) to find the optimal $G^*$. Having observed Theorem 3.1, Eq. (14) requires $p_{tu}$ to have a higher deviation from the average distribution, $p_{avg}$, while matching towards $p_s \approx p_{avg}$ and $p_{tk} \approx p_{avg}$. From this optimization procedure, we obtain the domain-invariant features over the source and the target-known domain, while segregating the target-unknowns, which is formalized as $p_s \approx p_{tk}$ and $p_{tu} \leftrightarrow \{p_{tk}, p_s\}$. We show that minimizing Eq. (14) does not lead to the negative infinity by the third term, regardless of the other KL Divergences, by Proposition 3.2.

**Proposition 3.2.** *(Proof in Appendix B.1.4) $D_{KL}(p_{tu} \| p_{avg})$ is bounded to $\log 2 - \log \lambda_{tu}$.*

This boundness guarantees that the first two KL Divergence terms maintain their influence on optimizing the parameter $\theta_g$ of $G$ stably while *segregating* the target-unknown features. Furthermore, we show Proposition 3.3 to characterize the feature alignment between $p_s$ and $p_{tk}$ under Eq. (14).

**Proposition 3.3.** *(Proof in Appendix B.1.5) Assume that $supp(p_s) \cap supp(p_{tu}) = \emptyset$ and $supp(p_{tk}) \cap supp(p_{tu}) = \emptyset$, where $supp(p) := \{z \in \mathcal{Z}|p(z) > 0\}$ is the support set of $p$. Then, the minimization w.r.t. $G$, by Eq. (14), is equivalent to the minimization of the summation on two $f$-divergences:*

$$D_{f_1}(p_s \| p_{tk}) + \lambda_{tk}D_{f_2}(p_{tk} \| p_s),$$

*where $f_1(u) = u \log \frac{u}{(1-\alpha)u+\alpha}$ and $f_2(u) = u \log \frac{u}{\alpha u+(1-\alpha)}$, with $\alpha = \frac{\lambda_{tk}}{1+\lambda_{tk}}$. Therefore, the minimum of Eq. (14) is achieved if and only if $p_s = p_{tk}$.*

Therefore, under the assumption, Proposition 3.3 theoretically guarantees that the source and the target-known feature distributions are aligned, which represents $p_s = p_{tk}$. From these analyses, we can theoretically guarantee both *alignment* and *segregation* from the proposed optimization.

## 3.4 Open-Set Recognition via Posterior Inference

This section starts from estimating $w_x = p(known|x)$ for a target instance, $x$. First, we utilize the labeled source dataset, $\chi_s$, to train the feature extractor $G$ and the open-set recognizer $E$, as below,

$$\mathcal{L}_e^s(\theta_g, \theta_e) = \frac{1}{n_s} \sum_{(x_s, y_s) \in \chi_s} \mathcal{L}_{ce}(E(G(x_s)), y_s), \tag{15}$$

where $E(G(x)) \in \mathbb{R}^{|\mathcal{C}_s|}$ contains the softmax activation. We note that the open-set recognizer, $E$, is a classifier to learn the decision boundary over $\mathcal{C}_s$ by the source domain.

Given the decision boundary by $E$, our assumption for open-set recognition includes two statements: for the target instances, 1) the higher entropy is caused by an uncertain classification case, and 2) the *unknown* class may induce the higher entropy because there are no prior training instances in the *unknown* classes. Based on two assumptions, the open-set recognizer, $E$, will provide higher entropy for the target-unknown instances than the target-known instances. Therefore, we consider the entropy value as the open-set recognition indicator. We verified our assumptions empirically in Figure 8a.

While we notice the entropy value from a single instance as the indicator between *known* and *unknown*, the open-set recognition should be holistically modeled over the target domain. Therefore, we model the mixture of two Beta distributions on the normalized target entropy values as below,

$$p(\ell_x) = \lambda_{tk} p(\ell_x | known) + \lambda_{tu} p(\ell_x | unknown), \qquad (16)$$

where the definition of $\lambda_{tk}$ and $\lambda_{tu}$ is the same as in the previous subsection. $\ell_x$ is the entropy value for the target instance, $x$, i.e., $\ell_x = H(E(G(x)))$ with the entropy function $H$. The Beta distribution is a perfect fit for this case of the closed interval support of $[0, 1]$, which is the normalized range of the entropy. Thus, we introduce the estimator $\hat{w}_x$ of $w_x$ by *Posterior Inference*, from fitting the Beta mixture model through the Expectation-Maximization (EM) algorithm (Details in Appendix B.2.1),

$$\hat{w}_x := p(known | \ell_x) = \frac{\lambda_{tk} p(\ell_x | known)}{\lambda_{tk} p(\ell_x | known) + \lambda_{tu} p(\ell_x | unknown)}, \qquad (17)$$

where the denominator is from Eq. (16). Since the fitting process incorporates all target information, the entire dataset is utilized to set a more informative weighting without any thresholds. Here, $\lambda_{tk}$ and $\lambda_{tu}$ are also explicitly estimated by the EM algorithm since they are not given (see Eq. (19) in Appendix B.2.1), whereas a threshold hyperparameter is used by the previous researches [19, 31].

### 3.5 Open-Set Classification

Given the proposed feature alignments, we train a classifier, $C$, to correctly classify the target instance over $\mathcal{C}_s$ or to reject as *unknown*. Note that the classifier, $C$, is the extended classifier with the dimension of $|\mathcal{C}_s| + 1$ to include the *unknown* class. Firstly, we construct the source classification loss on $\chi_s$. Secondly, for the target domain, we need to learn the decision boundary for the *unknown* class. Based on the proposed open-set recognition, $\hat{w}_x$, we train the classifier, $C$, as follows:

$$\mathcal{L}_{cls}(\theta_g, \theta_c) = \frac{1}{n_s} \sum_{(x_s, y_s) \in \chi_s} \mathcal{L}_{ce}(C(G(x_s)), y_s) + \frac{1}{n_t{}'} \sum_{x_t \in \chi_t} (1 - \hat{w}_x) \mathcal{L}_{ce}(C(G(x_t)), y_{unk}), \quad (18)$$

where $n_t{}'$ is a normalizing constant, and $y_{unk}$ is the one-hot vector for the *unknown* class. Furthermore, we incorporate entropy minimization loss for enforcing the unlabeled target instances to be recognized effectively [9, 13, 5], with the entropy loss function, $\mathcal{L}_H$,

$$\mathcal{L}_{ent}^t(\theta_g, \theta_c) = \frac{1}{n_t} \sum_{x_t \in \chi_t} \mathcal{L}_H(C(G(x_t))). \qquad (19)$$

### 3.6 Conditional UADAL

Domain adversarial learning may fail to capture a class-wise discriminative pattern in the features when the data distributions are complex [17]. Therefore, we also experimented with the conditional UADAL (cUADAL) with the discriminative information between classes. Inspired by [17], we replace the input information of the domain discriminator, $D$, from $G(x)$ to $[G(x), C(x)]$ in Eq. (6)-(7). cUADAL has the conditions on the prediction of the extended classifier, $C$, which is including the *unknown* dimension. Therefore, cUADAL provides more discriminative information than UADAL.

### 3.7 Training Procedure

We introduce how to update the networks, $E$, $G$, $D$, and $C$, based on the proposed losses. Before we start the training, we briefly fit the parameters of the posterior inference to obtain $\hat{w}_x$. This starting stage trains $E$ and $G$ on the source domain with a few iterations to catch the two modalities of the target entropy values, inspired by the previous work [1], as below,

$$(\hat{\theta}_e, \hat{\theta}_g) = \operatorname{argmin}_{\theta_g, \theta_e} \mathcal{L}_e^s(\theta_g, \theta_e). \qquad (20)$$

After the initial setup of $\hat{w}_x$, we train $D$, $C$, $E$, and $G$, alternatively. This alternation is composed of two phases: (i) learning $D$; and (ii) learning $G$, $C$, and $E$ networks. The first phase of the alternative iterations starts from learning the domain discriminator, $D$, by the optimization on $\mathcal{L}_D$ in Eq. (11),

$$\hat{\theta}_d = \operatorname{argmin}_{\theta_d} \mathcal{L}_D(\theta_g, \theta_d). \qquad (21)$$

Table 1 (HOS score %):

| Backbone (#) / Model | A-W | A-D | D-W | W-D | D-A | W-A | Avg. (Office-31) | P-R | P-C | P-A | A-P | A-R | A-C | R-A | R-P | R-C | C-R | C-A | C-P | Avg. (Office-Home) |
|---|---|---|---|---|---|---|---|---|---|---|---|---|---|---|---|---|---|---|---|---|
| **EfficientNet-B0 (5.3M)** | | | | | | | | | | | | | | | | | | | | |
| DANN | 63.2 | 72.7 | 92.6 | 94.8 | 63.7 | 57.2 | 74.0±0.3 | 35.7 | 16.5 | 18.2 | 34.1 | 46.3 | 22.9 | 40.7 | 47.8 | 28.2 | 12.4 | 7.5 | 13.4 | 27.0±0.3 |
| CDAN | 65.5 | 73.6 | 92.4 | 94.6 | 64.8 | 57.9 | 74.8±0.2 | 37.9 | 18.1 | 20.4 | 35.6 | 47.0 | 24.6 | 44.1 | 49.8 | 30.1 | 13.5 | 8.9 | 15.0 | 28.8±0.6 |
| STA | 58.3 | 62.2 | 81.6 | 79.6 | 69.8 | 67.4 | 69.8±1.2 | 59.4 | 43.6 | 51.9 | 53.8 | 60.6 | 49.5 | 58.8 | 53.5 | 49.9 | 53.4 | 49.5 | 49.4 | 52.8±0.2 |
| OSBP | 82.9 | 87.0 | 33.8 | 96.7 | 27.3 | 69.9 | 66.3±2.1 | 65.0 | 46.0 | 58.6 | 64.2 | 71.0 | 54.0 | 58.3 | 62.5 | 50.3 | 63.7 | 50.7 | 55.6 | 58.3±1.6 |
| ROS | 69.7 | 80.1 | 94.7 | 99.6 | 73.0 | 59.2 | 79.4±0.3 | 66.9 | 44.9 | 53.7 | 62.5 | 69.5 | 50.0 | 62.0 | 67.0 | 52.0 | 61.2 | 50.5 | 54.7 | 57.9±0.1 |
| DANCE | 68.1 | 68.8 | 91.3 | 85.0 | 68.5 | 63.3 | 74.2±4.0 | 17.2 | 47.5 | 7.2 | 26.6 | 19.6 | 36.6 | 2.2 | 19.8 | 10.9 | 6.4 | 4.3 | 19.0 | 18.1±2.9 |
| DCC | 87.2 | 69.1 | 89.4 | 94.4 | 63.5 | 76.1 | 79.9±2.9 | 72.2 | 41.0 | 56.5 | 66.4 | 75.7 | 52.8 | 55.9 | 71.5 | 49.9 | 60.4 | 48.1 | 60.8 | 59.3±1.5 |
| **UADAL** | **87.5** | **88.3** | **97.4** | 96.9 | **74.1** | 68.9 | **85.5±0.5** | **75.0** | 50.0 | 62.9 | **66.4** | **74.1** | 52.7 | 71.5 | 72.6 | 53.6 | 65.3 | 65.3 | **63.7** | 64.1±0.1 |
| **cUADAL** | 86.5 | **89.1** | 97.3 | **98.0** | 72.5 | **71.0** | **85.7±0.7** | 74.7 | **54.4** | **64.2** | 66.3 | 73.9 | 50.8 | 71.4 | **73.0** | 52.4 | 65.3 | 61.0 | 63.3 | **64.2±0.1** |
| **DenseNet-121 (7.9M)** | | | | | | | | | | | | | | | | | | | | |
| DANN | 71.9 | 72.0 | 90.2 | 85.3 | 73.8 | 72.3 | 77.6±0.5 | 68.8 | 35.4 | 48.7 | 62.6 | 71.9 | 45.3 | 62.8 | 68.7 | 45.9 | 62.2 | 47.0 | 54.7 | 56.2±0.3 |
| CDAN | 69.5 | 69.8 | 86.8 | 84.5 | 73.8 | 72.5 | 76.2±0.2 | 68.9 | 39.2 | 51.9 | 62.6 | 71.8 | 47.1 | 63.6 | 68.0 | 48.7 | 62.8 | 49.3 | 55.2 | 57.4±0.3 |
| STA | 77.0 | 68.6 | 84.0 | 77.2 | 76.6 | 75.1 | 76.4±1.5 | 65.6 | 46.1 | 58.4 | 55.8 | 64.3 | 50.4 | 62.6 | 58.6 | 51.1 | 61.0 | 56.0 | 55.9 | 57.1±0.1 |
| OSBP | 81.9 | 83.0 | 88.9 | 96.6 | 73.1 | 74.9 | 83.1±2.2 | 71.9 | 46.0 | 60.3 | 67.1 | 72.3 | 54.5 | 65.9 | 71.7 | 53.7 | 66.8 | 59.3 | 64.1 | 62.8±0.1 |
| ROS | 67.0 | 67.8 | 97.4 | 99.4 | 77.1 | 71.8 | 80.1±1.3 | 73.0 | 49.6 | 59.2 | 67.8 | 75.5 | 52.8 | 66.4 | 74.6 | 54.3 | 64.8 | 53.0 | 57.8 | 62.4±0.1 |
| DANCE | 69.9 | 67.8 | 84.0 | 82.8 | 79.9 | 81.1 | 77.6±0.3 | 51.8 | 51.0 | 59.7 | 63.9 | 58.2 | 58.2 | 43.4 | 48.9 | 55.0 | 41.3 | 54.6 | 60.6 | 53.9±0.5 |
| DCC | 83.9 | 80.8 | 88.4 | 93.1 | 79.7 | 80.4 | 84.4±1.3 | 75.1 | 46.6 | 58.0 | 70.8 | 78.6 | 56.6 | 63.4 | 75.5 | 55.8 | 71.3 | 55.0 | 63.3 | 64.2±0.2 |
| **UADAL** | **86.0** | 82.3 | **96.7** | 99.2 | 77.9 | 74.2 | **86.0±0.6** | **75.7** | 45.5 | 61.5 | **70.0** | **76.9** | 57.3 | 71.5 | 76.1 | **60.4** | **70.0** | 60.1 | 67.2 | 66.0±0.2 |
| **cUADAL** | 85.1 | **83.6** | 96.4 | 99.2 | 78.1 | 75.9 | **86.4±0.6** | 75.6 | 48.9 | **61.7** | **70.0** | 76.7 | **57.8** | **71.9** | **76.7** | 59.1 | 69.6 | 60.1 | **67.5** | **66.3±0.3** |
| **ResNet-50 (25.5M)** | | | | | | | | | | | | | | | | | | | | |
| DANN | 68.1 | 71.5 | 86.7 | 82.5 | 73.7 | 72.6 | 75.9±0.5 | 69.8 | 44.6 | 56.3 | 65.2 | 71.0 | 51.2 | 65.4 | 68.4 | 50.9 | 66.7 | 57.6 | 60.9 | 60.7±0.2 |
| CDAN | 64.9 | 66.8 | 84.3 | 80.5 | 72.7 | 71.0 | 73.4±1.3 | 69.7 | 47.2 | 58.6 | 65.1 | 70.7 | 52.9 | 66.0 | 67.6 | 52.7 | 67.1 | 58.2 | 61.7 | 61.4±0.3 |
| STA* | 75.9 | 75.0 | 69.8 | 75.2 | 73.2 | 66.1 | 72.5±0.8 | 69.5 | 53.2 | 61.9 | 54.0 | 68.3 | 55.8 | 67.1 | 64.5 | 54.5 | 66.8 | 57.4 | 60.4 | 61.1±0.3 |
| OSBP* | 82.7 | 82.4 | 97.2 | 91.1 | 75.1 | 73.7 | 83.7±0.4 | 73.9 | 53.2 | 63.2 | 65.2 | 72.9 | 55.1 | 66.7 | 72.3 | 54.5 | 70.6 | 64.3 | 64.7 | 64.7±0.2 |
| PGL* | 74.6 | 72.8 | 76.5 | 72.2 | 69.5 | 70.1 | 72.6±1.5 | 41.6 | 46.6 | 47.2 | 45.6 | 55.8 | 29.3 | 11.4 | 52.5 | 0.0 | 45.6 | 10.0 | 36.8 | 35.2 |
| ROS* | 82.1 | 82.4 | 96.0 | 99.7 | 77.9 | 77.2 | 85.9±0.2 | 74.4 | 56.3 | 60.6 | 69.3 | 76.5 | 60.1 | 68.8 | 75.7 | 60.4 | 68.6 | 58.9 | 65.2 | 66.2±0.3 |
| DANCE | 66.9 | 70.7 | 80.0 | 84.8 | 65.8 | 70.2 | 73.1±1.0 | 41.2 | 55.7 | 54.2 | 49.8 | 39.4 | 53.1 | 27.5 | 44.0 | 48.3 | 30.2 | 40.9 | 45.9 | 44.2±0.6 |
| DCC* | 87.1 | 85.5 | 91.2 | 87.1 | **85.5** | **84.4** | 86.8 | 64.0 | 52.8 | 59.5 | 67.4 | **80.6** | 52.9 | 56.0 | 62.7 | **76.9** | 67.0 | 49.8 | 66.6 | 64.2 |
| OSLPP* | 89.0 | **91.5** | 92.3 | 93.6 | 79.3 | 78.7 | 87.4 | 74.0 | **59.3** | 63.6 | 72.8 | 74.3 | 61.0 | 67.2 | 74.4 | 59.0 | 70.4 | 60.9 | 66.9 | 67.0 |
| **UADAL** | 89.1 | 86.0 | 97.8 | 99.5 | 79.7 | 76.5 | 88.1±0.2 | **76.9** | 56.6 | 63.9 | 70.8 | 77.4 | 63.2 | 72.1 | **76.8** | 60.6 | **73.4** | 64.2 | **69.5** | **68.7±0.2** |
| **cUADAL** | **90.1** | 87.9 | **98.2** | 99.4 | 80.5 | 75.1 | **88.5±0.3** | 76.8 | 54.6 | 62.9 | 71.6 | 77.5 | 63.6 | 72.6 | 76.7 | 59.9 | 72.6 | **65.0** | 68.3 | 68.5±0.1 |

Table 1: HOS score (%) on Office-31 & Office-Home using EfficientNet-B0 (5.3M), DenseNet-121 (7.9M), and ResNet-50 (25.5M) as backbone network, where (#) represents the number of the parameters. A-W in a column means that A is the source domain and W is the target domain. **Avg.** is the averaged HOS over all tasks in each dataset. (bold: best performer, underline: second-best performer, *: officially reported performances.) The detailed experimental results including the other metrics such as OS* and UNK are in Appendix C.2.9

Based on the updated $\hat{\theta}_d$, the parameter, $\theta_g$, is associated with both optimization loss and classification loss. Therefore, for the second phase, the parameters, $\theta_g$ and $\theta_c$, are optimized as below,

$$(\hat{\theta}_g, \hat{\theta}_c) = \operatorname{argmin}_{\theta_g, \theta_c} \mathcal{L}_{cls}(\theta_g, \theta_c) + \mathcal{L}_{ent}^t(\theta_g, \theta_c) - \mathcal{L}_G(\theta_g, \hat{\theta}_d). \quad (22)$$

We also keep updating $\theta_e$ by Eq. (20), which leads to a better fitting of the posterior inference on the aligned features (empirically shown in Figure 9b). Algorithm 1 in Appendix B.3.1 enumerates the detailed training procedures of UADAL. **Computational Complexity** is asymptotically the same as existing domain adversarial learning methods, which linearly grow by the number of instances $O(n)$. The additional complexity may come from the posterior inference requiring $O(nk)$; $k$ is a constant number of EM iterations (the detailed analysis on the computations in Appendix B.3.2 and C.2.8).

## 4 Experiments

### 4.1 Experimental Settings

**Datasets** We utilized several benchmark datasets. **Office-31** [22] consists of three domains: Amazon (A), Webcam (W), and DSLR (D) with 31 classes. **Office-Home** [30] is a more challenging dataset with four different domains: Artistic (A), Clipart (C), Product (P), and Real-World (R), containing 65 classes. **VisDA** [21] is a large-scale dataset from synthetic images to real one, with 12 classes. In terms of the class settings, we follow the experimental protocols by [23].

**Baselines** We compared UADAL and cUADAL with several baselines by choosing the recent works in CDA, OSDA, and Universal DA (UniDA). For CDA, we choose DANN [7] and CDAN [17] to show that the existing domain adversarial learning for CDA is not suitable for OSDA. For OSDA, we compare UADAL to OSBP [23], STA [13], PGL [19], ROS [5], and OSLPP [31]. For UniDA, we conduct the OSDA setting in DANCE [24] and DCC [12].

**Implementation** We commonly utilize three alternative pre-trained backbones: 1) ResNet-50 [10], 2) DenseNet-121 [11], and 3) EfficientNet-B0 [26] to show the robustness on backbone network choices, while using VGGNet [25] on VisDA for a fair comparison. (Details in Appendix C.1).

**Metric** We utilize HOS, which is a harmonic mean of OS*, and UNK [5]. OS* is a class-wise averaged accuracy over *known* classes, and UNK is an accuracy only for the *unknown* class. HOS is suitable for OSDA because it is higher when performing well in both *known* and *unknown* classifications.

| Method | EfficientNet-B0 | | | DenseNet-121 | | | ResNet-50 | | | VGGNet | | |
|---|---|---|---|---|---|---|---|---|---|---|---|---|
| | OS* | UNK | HOS | OS* | UNK | HOS | OS* | UNK | HOS | OS* | UNK | HOS |
| STA | 49.3 | 56.4 | 52.5 | 53.1 | 76.7 | 62.7 | 56.9 | 75.8 | 65.0 | 63.9* | 84.2* | 72.7* |
| OSBP | 48.8 | 70.4 | 57.6 | 48.7 | 65.7 | 55.9 | 50.0 | 77.2 | 60.7 | 59.2* | 85.1* | 69.8* |
| PGL | 56.9 | 26.1 | 35.8 | - | - | - | 70.3 | 33.4 | 45.3 | 82.8* | 68.1* | 74.7* |
| ROS | 36.7 | 72.0 | 48.7 | 42.7 | 81.1 | 55.9 | 45.8 | 64.8 | 53.7 | - | - | - |
| DANCE | 38.9 | 63.2 | 48.1 | 59.8 | 67.3 | 62.3 | 61.3 | 72.9 | 66.5 | - | - | - |
| DCC | 38.3 | 54.2 | 44.8 | 16.7 | 76.9 | 27.2 | 13.4 | 88.0 | 23.3 | 68.0* | 73.6* | 70.7* |
| OSLPP | - | - | - | - | - | - | - | - | - | - | - | - |
| UADAL | 47.0 | 76.8 | 58.3 | 56.2 | 81.5 | 66.5 | 58.0 | 86.2 | 69.4 | 63.1 | 93.3 | 75.3 |
| cUADAL | 47.2 | 76.5 | 58.4 | 57.9 | 84.1 | 68.6 | 58.5 | 87.6 | 70.1 | 64.3 | 92.6 | 75.9 |

Table 2: Results on VisDA dataset. (bold/underline/*: Refer to the Table 1)

| Office-31 with ResNet-50 | | |
|---|---|---|
| Weight | Core | HOS Avg. |
| STA | STA | 72.5±0.8 |
| | UADAL | 85.5±0.9 |
| ROS | ROS | 85.9±0.2 |
| | UADAL | 86.7±0.2 |
| UADAL | UADAL | 88.1±0.2 |

Table 3: Ablation Studies for Different Weighting / Core Schemes on STA and ROS.

## 4.2 Experimental Results

**Quantitative Analysis** Table 1 reports the quantitative performances of UADAL, cUADAL, and baselines applied to Office-31 and Office-Home, with three backbone networks. UADAL and cUADAL show statistically significant improvements compared to *all* existing baselines in both Office-31 and Office-Home. Moreover, there is a large improvement by UADAL from baselines in EfficientNet-B0 and DenseNet-121, even with fewer parameters. In terms of the table on ResNet-50, for a fair comparison, UADAL outperforms the most competitive model, OSLPP (HOS=67.0). Compared to the domain adversarial learning methods, such as STA and OSBP, UADAL significantly outperforms in all cases. UADAL also performs better than other self-supervised learning methods such as ROS and DCC. These performance increments are commonly observed in all three backbone structures and in two benchmark datasets, which represents the effectiveness of the proposed model.

Table 2 shows the experimental results on VisDA, which is known to have a large domain shift from a synthetic to a real domain. It demonstrates that UADAL and cUADAL outperform the other baselines [‡] under the significant domain shift. Moreover, the baselines with the feature alignments by the domain adversarial learning, such as STA, OSBP, and PGL, outperform the self-supervised learning models, ROS, DANCE, and DCC, in most cases. Therefore, these results support that OSDA essentially needs the feature alignments when there are significant domain shifts. The detailed discussions about the low accuracies of some baselines are in Appendix C.2.2. In addition, we observe that cUADAL mostly performs better than UADAL, which shows that cUADAL provides more discriminative information both on *known* and *unknown* by the prediction of the classifier $C$.

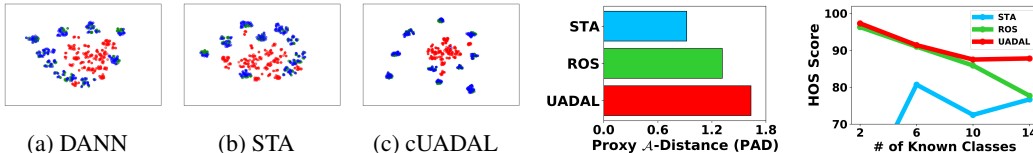

(a) DANN    (b) STA    (c) cUADAL

Figure 3: t-SNE Visualization on D → W of Office-31. (Blue/Red/Green: Target-Known/-Unknown/Source)

Figure 4: PAD value on *tk/tu* feature distributions.

Figure 5: Openness on Office-31 (ResNet-50).

**Qualitative Analysis** Figure 3 represents the t-SNE visualizations [29] of the learned features by ResNet-50 (the details in Appendix C.2.3). We observe that the target-unknown (red) features from the baselines are not segregated from the source (green) and the target-known (blue) features. On the contrary, UADAL aligns the features from the source and the target-known instances accurately with clear *segregation* of the target-unknown features. In order to investigate the learned feature distributions quantitatively, Figure 4 provides Proxy $\mathcal{A}$-Distance (PAD) between the feature distributions from STA, ROS, and UADAL. PAD is an empirical measure of distance between domain distributions [8] (Details in Appendix C.2.4), and a higher PAD means clear discrimination between two distributions. Thus, we calculate PAD between the target-known ($tk$) and the target-unknown ($tu$) feature distributions from the feature extractor, $G$. We found that PAD from UADAL is much higher than STA and ROS. It shows that UADAL explicitly segregates the unknown features.

In order to provide more insights and explanations compared to STA, we provide an analysis of the correlation between the evaluation metric and the distance measure of the feature distributions. We utilize UNK, and HOS as evaluation metrics, and PAD between $tk$ and $tu$ as the distance measure. Figure 6 shows the scatter plots of (UNK & PAD) and (HOS & PAD). The grey arrows mean the corresponding tasks by UADAL and STA. As shown in the figure, HOS and UNK have a positive correlation with the distance between the target-unknown ($tu$) and the target-known ($tk$). In simple

---

[‡]OSLPP is infeasible to a large dataset, such as VisDA dataset (Detailed discussions in Appendix C.2.1).

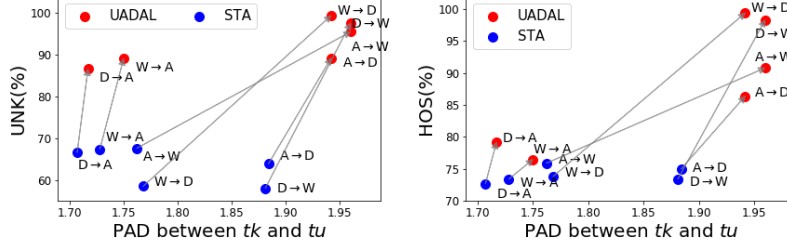

Figure 6: Correlation analysis between the PAD value and each of UNK (left) and HOS (right), in Office-31 dataset (with ResNet-50). The arrow represents the same task, such as A→W, for UADAL and STA.

words, better segregation enables better HOS and UNK. Specifically, from STA to UADAL on the same task, the distances (PAD on $tk/tu$) are increased, and the corresponding performances, UNK and HOS, are also increased. It means that UADAL effectively segregates the target feature distributions, and then leads to better performance for OSDA. In other words, the proposed unknown-aware feature alignment is important to solve the OSDA problem. From this analysis, our explicit segregation of $tu$ is a key difference from the previous methods. Therefore, we claimed that this explicit segregation of UADAL is the essential part of OSDA, and leads to better performances.

**Robust on Openness** We conduct the openness experiment to show the robustness of the varying number of the known classes. Figure 5 represents that UADAL performs better than the baselines over all cases. Our *open-set recognition* does not need to set any thresholds since we utilize the loss information of the target domain, holistically. Therefore, UADAL is robust to the degree of openness.

### 4.3 Ablation Studies

**Effectiveness of Unknown-Aware Feature Alignment** We conducted ablation studies to investigate whether the unknown-aware feature alignment is essential for the OSDA problem.

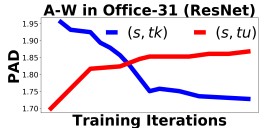

Table 3 represents the experimental results where the feature alignment procedure of STA or ROS is replaced by UADAL, maintaining their weighting schemes, in Office-31 with ResNet-50. In terms of each weighting, the UADAL core leads to better performances in both STA and ROS cases. Surprisingly, when we replace STA's known-only matching with the unknown-aware feature alignment of UADAL, the performance increases dramatically. It means that the implicit separation of STA at the classification level is limited to segregating the target-unknown features.

Figure 7: Convergence of Sequential Optimization.

Therefore, explicit *segregation* by UADAL is an essential part to solve the OSDA problem. Moreover, Figure 7 shows the empirical convergence of the proposed sequential optimization as PAD value between $p_s$ and $p_{tk}$ and between $p_s$ and $p_{tu}$ over training, i.e., $p_s \approx p_{tk}$ and $p_{tu} \leftrightarrow \{p_{tk}, p_s\}$.

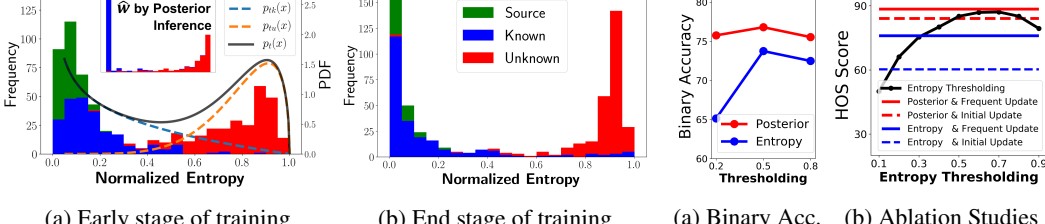

(a) Early stage of training     (b) End stage of training     (a) Binary Acc.    (b) Ablation Studies

Figure 8: (a) The histogram of the source and target entropy with the fitted posterior inference model, at the Early / End stage of training, where the subfigure in (a) shows the histogram of $\hat{w}_x$ at this stage.

Figure 9: (a) Binary accuracy on $tk/tu$ by thresholding $\hat{w}_x$. (b) Ablations for posterior inference with thresholding.

**Superiorities of Open-set Recognition:** *(Effective Recognition)* Figure 8a represents the histogram of the source/target entropy values from $E$ after the early stage of training by Eq. (20) (sensitivity analysis on the varying number of iterations for the early stage in Appendix C.2.5). Therefore, it demonstrates that our assumption on two modalities of the entropy values holds, robustly. Furthermore, the subfigure in Figure 8a shows the histogram of the $\hat{w}_x$ by the posterior inference. It shows that our *open-set recognition*, $\hat{w}_x$, provides a more clear separation for the target-known (blue) and

the target-unknown (red) instances than the entropy values. Figure 8b represents the histogram of the entropy at the end of the training, which is further split than Figure 8a. This clear separation implicitly supports that the proposed feature alignment accomplishes both *alignment* and *segregation*.

*(Informativeness)* Furthermore, we conduct a binary classification task by thresholding the weights, $\hat{w}_x$, whether it predicts $tk$ or $tu$ correctly. Figure 9a represents the accuracies over the threshold values when applying the posterior inference or the normalized entropy values themselves, in Office-31. It shows that the posterior inference discriminates $tk$ and $tu$ more correctly than using the entropy value itself. Additionally, the posterior inference has similar accuracies over the threshold values (0.2, 0.5, and 0.8), which also indicates a clear separation. Therefore, we conclude that the posterior inference provides a more informative weighting procedure than the entropy value itself.

*(Comparison to Thresholding)* Figure 9b shows that UADAL with the posterior inference outperforms all cases of thresholding the entropy values. It means that the posterior inference is free to set thresholds. Moreover, the results demonstrate that UADAL with the posterior inference outperforms the case of using the normalized entropy value, and its frequent updating (solid line) leads to a better performance than only initial updating (dashed line). It is consistent with the analysis from Figure 8b.

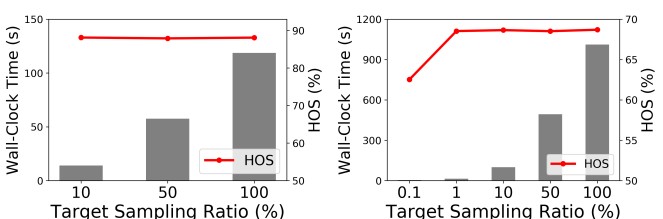

Figure 10: Ablation study of applying sampling on the target domain with Office-31 (left) and Office-Home (right) w.r.t. the averaged scores.

| Dataset | Ent. | OS* | UNK | HOS |
|---|---|---|---|---|
| Office-31 | X | 85.4 | 90.0 | 87.5 |
| | O | 84.8 | 82.1 | **88.1** |
| Office-Home | X | 61.9 | 75.2 | 67.6 |
| | O | 62.6 | 78.0 | **68.7** |

Table 4: Ablation on Entropy Minimization (Ent.) Loss from Eq. (19) on the Office-31 and Office-Home. The values in the table are the averaged score over the tasks in each dataset. The detailed results are in Table 2 in Appendix C.2.7.

*(Efficiency)* In terms of computation, the posterior inference increases the complexity because we need to fit the mixture model, where the computational complexity is $O(nk)$ with the number of samples, $n$, and the number of fitting iterations, $k$ ($n \gg k$). As an alternative, we fit the mixture model only by sampling the target instance. Figure 10 represents the wall-clock time increased by the fitting process and the performances over the sampling ratio (%) for the target domain. Since the computational complexity is linearly increased by the number of samples, the wall-clock time is also linearly increased by increasing the sampling ratio. Interestingly, we observed that even when the sampling ratio is small, i.e. 10%, the performances do not decrease, on both Office-31 and Office-Home datasets. The qualitative analysis of this sampling is provided in Appendix C.2.8.

**Entropy Minimization Loss** In order to show the effect of the entropy minimization by Eq. (19), we conduct the ablation study on the Office-31 and Office-Home. Table 4 shows that the entropy minimization leads to better performance. However, UADAL without the loss still performs better than the other baselines (compared with Table 1). It represents that UADAL learns the feature space appropriately as we intended to suit Open-Set Domain Adaptation. The properly learned feature space leads to effectively classifying the target instances without entropy minimization.

## 5   Conclusion

We propose Unknown-Aware Domain Adversarial Learning (UADAL), which is the first approach to explicitly design the *segregation* of the unknown features in the domain adversarial learning for OSDA. We design a new domain discrimination loss and formulate the sequential optimization for the unknown-aware feature alignment. Empirically, UADAL outperforms all baselines on all benchmark datasets with various backbone networks, by reporting state-of-the-art performances.

## Acknowledgments and Disclosure of Funding

This research was supported by Research and Development on Key Supply Network Identification, Threat Analyses, Alternative Recommendation from Natural Language Data (NRF) funded by the Ministry of Education (2021R1A2C200981612).

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
