# Appendix:
# Unknown-Aware Domain Adversarial Learning for Open-Set Domain Adaptation

First, Section A provides comprehensive literature reviews. Second, Section B provides the details for the proposed model, UADAL. It consists of three parts; 1) Sequential Optimization Problem (Section B.1), 2) Posterior Inference (Section B.2), and 3) Training Details (Section B.3). Third, Section C provides the experimental details, including the implementation details (Section C.1) and the detailed experimental results (Section C.2). Finally, Section D shows 'Limitations and Potential Negative Societal Impacts' of this work.

Note that all references and the equation numbers are independent of the main paper. We utilize the expression of "Eq. (XX) in the main paper", especially when referring the equations from the main paper.

## A    Literature Reviews

**(Closed-Set) Domain Adaptation (DA)** is the task of leveraging the knowledge of the labeled source domain to the target domain [5, 3, 2, 22, 11]. Here, DA assumes that the class sets from the source and the target domain are identical. The typical approaches to solve DA have focused on minimizing the discrepancy between the source and the target domain since they are assumed to be drawn from the different distributions [31, 41, 23, 24]. The discrepancy-based approaches have been proposed to define the metric to measure the distance between the source and the target domain in the feature space [22, 46]. Other works are based on adversarial methods [10, 42, 25]. These approaches introduce the domain discriminator and a feature generator in order to learn the feature space to be domain-invariant. Self-training methods are also proposed to mitigate the DA problems by pseudo-labeling the target instances [21, 18, 28, 32], originally tailored to semi-supervised learning [13, 38].

**Open-Set Recognition (OSR)** is the task to classify an instance correctly if it belongs to the known classes or to reject outliers otherwise, during the testing phase [35]. Here, the outliers are called 'open-set' which is not available in the training phase. Many approaches have been proposed to solve the OSR problem [4, 29, 39, 30, 36, 7, 8]. OpenMax [4] is the first deep learning approach for OSR, introducing a new model layer to estimate the probability of an instance being *unknown* class, based on the Extreme Value Theory (EVT). OSRCI [29], another stream of the approaches utilizing GANs, generates the virtual images which are similar to the training instances but do not belong to the known classes. Other approaches contain the reconstruction-based methods [39, 30] and prototype-based methods [36, 7, 8]. Also, [43] claims that the performance of OSR is highly correlated with its accuracy on the closed-set classes. This claim is associated with our open-set recognizer.

**Open-Set Domain Adaptation** is a more realistic and challenging task of Domain Adaptation, where the target domain includes the open-set instances which are not discovered by the source domain. In terms of the domain adversarial learning, in addition to STA, OSBP [33] utilizes a classifier to predict the target instances to the pre-determined threshold, and trains the feature extractor to deceive the classifier for aligning to *known* classes or rejecting as *unknown* class. However, their recognition on *unknown* class only depends on the threshold value, without considering the data instances. PGL [27] introduces progressive graph-based learning to regularize the class-specific manifold, while jointly

optimizing with domain adversarial learning. However, their adversarial loss includes all instances of the target domain, which is critically weakened by the negative transfer.

In terms of the self-supervised learning, ROS [6] is rotation-based self-supervised learning to compute the normality score for separating target known/unknown information. DANCE [34] is based on a self-supervised clustering to move a target instance either to shared-known prototypes in the source domain or to its neighbor in the target domain. DCC [19] is a domain consensus clustering to exploit the intrinsic structure of the target domain. However, these approaches do not have any feature alignments between the source and the target domain, which leads to performance degradation under the significant domain shifts.

There is also a notable work, OSLPP [44] optimizing projection matrix toward a common subspace to class-wisely align the source and the target domain. This class-wise matching depends on the pseudo label for the target instances. However, inaccurate pseudo labels such as early mistakes can result in error accumulation and domain misalignment [32]. Moreover, the optimization requires the pair-wise distance calculation, which results in a growing complexity of $O(n^2)$ by the $n$ data instances. It could be limited to the large-scaled domain.

# B  Algorithm and Optimization Details

## B.1  Sequential Optimization Problem

### B.1.1  Decomposition of Domain Discrimination Loss

The domain discrimination loss for the target domain is as below, (Eq. (7) in the main paper)

$$\mathcal{L}_d^t(\theta_g, \theta_d) = \mathbb{E}_{p_t(x)}\left[ -w_x \log D_{tk}(G(x)) - (1 - w_x) \log D_{tu}(G(x)) \right],$$

where $w_x = p(known|x)$ is the probability of a target instance, $x$, belonging to a target-known class. Then, we decompose $\mathcal{L}_d^t(\theta_g, \theta_d)$ into the two terms, $\mathcal{L}_d^{tk}(\theta_g, \theta_d)$ and $\mathcal{L}_d^{tu}(\theta_g, \theta_d)$, as follows,

$$\mathcal{L}_d^t(\theta_g, \theta_d) = \mathcal{L}_d^{tk}(\theta_g, \theta_d) + \mathcal{L}_d^{tu}(\theta_g, \theta_d),$$
$$\mathcal{L}_d^{tk}(\theta_g, \theta_d) := \lambda_{tk} \cdot \mathbb{E}_{p_{tk}(x)}\left[ -\log D_{tk}(G(x)) \right],$$
$$\mathcal{L}_d^{tu}(\theta_g, \theta_d) := \lambda_{tu} \cdot \mathbb{E}_{p_{tu}(x)}\left[ -\log D_{tu}(G(x)) \right],$$

where $p_{tk}(x) := p_t(x|known)$ and $p_{tu}(x) := p_t(x|unknown)$; $\lambda_{tk} = p(known)$; and $\lambda_{tu} = p(unknown)$.

*Proof.* We start this proof from Eq. (7) in the main paper,

$$\mathcal{L}_d^t(\theta_g, \theta_d) = \mathbb{E}_{p_t(x)}\left[ -w_x \log D_{tk}(G(x)) - (1 - w_x) \log D_{tu}(G(x)) \right].$$

For the convenience of the derivation, we replace $w_x$ as $p(known|x)$.

$$\mathcal{L}_d^t(\theta_g, \theta_d) = \mathbb{E}_{x \sim p_t(x)}\left[ -p_t(known|x) \log D_{tk}(G(x)) - p_t(unknown|x) \log D_{tu}(G(x)) \right]$$

$$= \int\limits_{x \sim p_t(x)} \Big( -p_t(known|x) \log D_{tk}(G(x)) - p_t(unknown|x) \log D_{tu}(G(x)) \Big) dx$$

$$= \int\limits_{x} \Big( -p_t(x)(p_t(known|x) \log D_{tk}(G(x)) - p_t(x)p_t(unknown|x) \log D_{tu}(G(x)) \Big) dx$$

$$= \int\limits_{x} \Big( -p_t(known, x) \log D_{tk}(G(x)) - p_t(unknown, x) \log D_{tu}(G(x)) \Big) dx$$

$$= \int\limits_{x} -p_t(x|known)p_t(known) \log D_{tk}(G(x)) \, dx + \int\limits_{x} -p_t(x|unknown)p_t(unknown) \log D_{tu}(G(x)) \, dx$$

$$= p_t(known) \int\limits_{x} -p_t(x|known) \log D_{tk}(G(x)) \, dx + p_t(unknown) \int\limits_{x} -p_t(x|unknown) \log D_{tu}(G(x)) \, dx$$

$$= p_t(known) \int\limits_{x \sim p_{tk}(x)} -\log D_{tk}(G(x)) \, dx + p_t(unknown) \int\limits_{x \sim p_{tu}(x)} -\log D_{tu}(G(x)) \, dx$$

$$= p_t(known)\mathbb{E}_{x \sim p_{tk}(x)} \left[ -\log D_{tk}(G(x)) \right] + p_t(unknown)\mathbb{E}_{x \sim p_{tu}(x)} \left[ -\log D_{tu}(G(x)) \right]$$

$$= \lambda_{tk}\mathbb{E}_{x \sim p_{tk}(x)} \left[ -\log D_{tk}(G(x)) \right] + \lambda_{tu}\mathbb{E}_{x \sim p_{tu}(x)} \left[ -\log D_{tu}(G(x)) \right].$$

Thus, we define new terms with respect to $p_{tk}(x)$ and $p_{tu}(x)$ as follow.

$$\mathcal{L}_d^{tk}(\theta_g, \theta_d) := \lambda_{tk}\mathbb{E}_{x \sim p_{tk}(x)} \left[ -\log D_{tk}(G(x)) \right]$$
$$\mathcal{L}_d^{tu}(\theta_g, \theta_d) := \lambda_{tu}\mathbb{E}_{x \sim p_{tu}(x)} \left[ -\log D_{tu}(G(x)) \right]$$

Therefore, by the above derivation, we decompose $\mathcal{L}_d^t(\theta_g, \theta_d)$ as follow,

$$\mathcal{L}_d^t(\theta_g, \theta_d) = \mathcal{L}_d^{tk}(\theta_g, \theta_d) + \mathcal{L}_d^{tu}(\theta_g, \theta_d).$$

$\square$

### B.1.2  Optimal point of the domain discriminator $D$

The optimal $D^*$ given the fixed $G$ is as follow (Eq. (13) in the main paper),

$$D^*(G(x; \theta_g)) = D^*(z) = \left[ \frac{p_s(z)}{2p_{avg}(z)}, \frac{\lambda_{tk}p_{tk}(z)}{2p_{avg}(z)}, \frac{\lambda_{tu}p_{tu}(z)}{2p_{avg}(z)} \right],$$

where $p_{avg}(z) = (p_s(z) + \lambda_{tk}p_{tk}(z) + \lambda_{tu}p_{tu}(z))/2$. Note that $z \in \mathcal{Z}$ stands for the feature space from $G$. In other words, $p_d(z) = \{G(x; \theta_g)|x \sim p_d(x)\}$ where $d$ is $s$, $tk$, or $tu$.

*Proof.* First, we fix $G$, and optimize the problem with respect to $D$.

$$\min_{\theta_d} \mathcal{L}_D(\theta_g, \theta_d) = \mathcal{L}_d^s(\theta_g, \theta_d) + \mathcal{L}_d^{tk}(\theta_g, \theta_d) + \mathcal{L}_d^{tu}(\theta_g, \theta_d)$$

$$= - \int\limits_{x \sim p_s(x)} \log D_s(G(x)) \, dx - \lambda_{tk} \int\limits_{x \sim p_{tk}(x)} \log D_{tk}(G(x)) \, dx - \lambda_{tu} \int\limits_{x \sim p_{tu}(x)} \log D_{tu}(G(x)) \, dx$$

$$= - \int\limits_{z \sim p_s(z)} \log D_s(z) \, dz - \lambda_{tk} \int\limits_{z \sim p_{tk}(z)} \log D_{tk}(z) \, dz - \lambda_{tu} \int\limits_{x \sim p_{tu}(z)} \log D_{tu}(z) \, dz$$

$$= \int\limits_{z} \Big( -p_s(z) \log D_s(z) - \lambda_{tk}p_{tk}(z) \log D_{tk}(z) - \lambda_{tu}p_{tu}(z) \log D_{tu}(z) \Big) dz$$

Also, note that $D_s(z) + D_{tk}(z) + D_{tu}(z) = 1$ for all $z$. Therefore, we transform the optimization problem as follow [12]:

$$\min_{\theta_d} \quad -p_s(z) \log D_s(z) - \lambda_{tk}p_{tk}(z) \log D_{tk}(z) - \lambda_{tu}p_{tu}(z) \log D_{tu}(z)$$

$$\text{s.t.} \quad D_s(z) + D_{tk}(z) + D_{tu}(z) = 1$$

for all $z$. We introduce the Lagrange variable $v$ to use Lagrange multiplier method.

$$\min_{\theta_d} \tilde{\mathcal{L}}_D := - p_s(z) \log D_s(z) - \lambda_{tk} p_{tk}(z) \log D_{tk}(z) - \lambda_{tu} p_{tu}(z) \log D_{tu}(z)$$
$$+ v(D_s(z) + D_{tk}(z) + D_{tu}(z) - 1)$$

To find optimal $D^*$, we find the derivative of $\tilde{\mathcal{L}}_D$ with respect to $D$ and $v$.

$$\frac{\partial \tilde{\mathcal{L}}_D}{\partial D_s(z)} = \frac{-p_s(z)}{D_s(z)} + v = 0 \quad \Leftrightarrow \quad D_s(z) = \frac{p_s(z)}{v}$$

$$\frac{\partial \tilde{\mathcal{L}}_D}{\partial D_{tk}(z)} = \frac{-\lambda_{tk} p_{tk}(z)}{D_{tk}(z)} + v = 0 \quad \Leftrightarrow \quad D_{tk}(z) = \frac{\lambda_{tk} p_{tk}(z)}{v}$$

$$\frac{\partial \tilde{\mathcal{L}}_D}{\partial D_{tu}(z)} = \frac{-\lambda_{tu} p_{tu}(z)}{D_{tu}(z)} + v = 0 \quad \Leftrightarrow \quad D_{tu}(z) = \frac{\lambda_{tu} p_{tu}(z)}{v}$$

$$\frac{\partial \tilde{\mathcal{L}}_D}{\partial v} = D_s(z) + D_{tk}(z) + D_{tu}(z) - 1 = 0 \quad \Leftrightarrow \quad D_s(z) + D_{tk}(z) + D_{tu}(z) = 1$$

From the above equations, we have

$$D_s(z) + D_{tk}(z) + D_{tu}(z) = \frac{p_s(z)}{v} + \frac{\lambda_{tk} p_{tk}(z)}{v} + \frac{\lambda_{tu} p_{tu}(z)}{v} = 1,$$

then,

$$v = p_s(z) + \lambda_{tk} p_{tk}(z) + \lambda_{tu} p_{tu}(z) = 2p_{avg}(z).$$

Thus, we get optimal $D^*$ as

$$D^*(z) = [D_s^*(z), D_{tk}^*(z), D_{tu}^*(z)] = \left[ \frac{p_s(z)}{2p_{avg}(z)}, \frac{\lambda_{tk} p_{tk}(z)}{2p_{avg}(z)}, \frac{\lambda_{tu} p_{tu}(z)}{2p_{avg}(z)} \right].$$

$\square$

### B.1.3   Proof of Theorem 3.1

**Theorem B.1.** *Let $\theta_d^*$ be the optimal parameter of $D$ by optimizing Eq. (11) in the main paper. Then, $-\mathcal{L}_G(\theta_g, \theta_d^*)$ can be expressed as, with a constant $C_0$,*
$$-\mathcal{L}_G(\theta_g, \theta_d^*) = D_{KL}(p_s \| p_{avg}) + \lambda_{tk} D_{KL}(p_{tk} \| p_{avg}) - \lambda_{tu} D_{KL}(p_{tu} \| p_{avg}) + C_0.$$

*Proof.* First, we change maximization problem into minimization problem, and substitute $D^*$ by using Eq. (13) in the main paper. Note that $p_{avg} = (p_s(z) + \lambda_{tk} p_{tk}(z) + \lambda_{tu} p_{tu}(z))/2$.

$$\min_{\theta_g} -\mathcal{L}_G(\theta_g, \theta_d) = -\mathcal{L}_d^s(\theta_g, \theta_d) - \mathcal{L}_d^{tk}(\theta_g, \theta_d) + \mathcal{L}_d^{tu}(\theta_g, \theta_d) \tag{1}$$

$$= \int_z (p_s(z) \log D_s^*(z) + \lambda_{tk} p_{tk}(z) \log D_{tk}^*(z) - \lambda_{tu} p_{tu}(z) \log D_{tu}^*(z)) \, dz \tag{2}$$

$$= \int_z \left( p_s(z) \log \frac{p_s(z)}{2p_{avg}(z)} + \lambda_{tk} p_{tk}(z) \log \frac{\lambda_{tk} p_{tk}(z)}{2p_{avg}(z)} - \lambda_{tu} p_{tu}(z) \log \frac{\lambda_{tu} p_{tu}(z)}{2p_{avg}(z)} \right) dz \tag{3}$$

$$= D_{KL}(p_s \| p_{avg}) + \lambda_{tk} D_{KL}(p_{tk} \| p_{avg}) - \lambda_{tu} D_{KL}(p_{tu} \| p_{avg}) + C_0 \tag{4}$$

where $C_0 = -2\lambda_{tk} \log 2 + \lambda_{tk} \log \lambda_{tk} - \lambda_{tu} \log \lambda_{tu}$.

We use below derivation for the last equation, i.e. from Eq. (3) to Eq. (4).

$$D_{KL}(p_s \| p_{avg}) = \int_z p_s(z) \log \frac{p_s(z)}{(p_{avg}(z))} \, dz$$

$$= \int_z p_s(z) \log \frac{2p_s(z)}{p_s(z) + \lambda_{tk} p_{tk}(z) + \lambda_{tu} p_{tu}(z)} \, dz$$

$$= \int_z p_s(z) \log \frac{p_s(z)}{p_s(z) + \lambda_{tk} p_{tk}(z) + \lambda_{tu} p_{tu}(z)} \, dz$$

$$+ \int_z p_s(z) \log 2 \, dz$$

$$= \int_z p_s(z) \log \frac{p_s(z)}{2p_{avg}(z)} \, dz + \int_z p_s(z) \log 2 \, dz$$

Thus,

$$\int_z p_s(z) \log \frac{p_s(z)}{2p_{avg}(z)} \, dz = D_{KL}(p_s \| p_{avg}) - \log 2 \tag{5}$$

For the second term,

$$
\begin{aligned}
&D_{KL}(p_{tk} \| p_{avg}) \\
&= \int_z p_{tk}(z) \log \frac{p_{tk}(z)}{p_{avg}(z)} \, dz \\
&= \int_z p_{tk}(z) \log \frac{2p_{tk}(z)}{p_s(z) + \lambda_{tk} p_{tk}(z) + \lambda_{tu} p_{tu}(z)} \, dz \\
&= \int_z p_{tk}(z) \log \frac{p_{tk}(z)}{p_s(z) + \lambda_{tk} p_{tk}(z) + \lambda_{tu} p_{tu}(z)} \, dz + \int_z p_{tk}(z) \log 2 \, dz \\
&= \int_z p_{tk}(z) \log \frac{p_{tk}(z)}{p_s(z) + \lambda_{tk} p_{tk}(z) + \lambda_{tu} p_{tu}(z)} \, dz + \log 2 \\
&= \int_z p_{tk}(z) (\log \frac{p_{tk}(z)}{p_s(z) + \lambda_{tk} p_{tk}(z) + \lambda_{tu} p_{tu}(z)} + \log \lambda_{tk} - \log \lambda_{tk}) \, dz + \log 2 \\
&= \int_z p_{tk}(z) \log \frac{\lambda_{tk} p_{tk}(z)}{p_s(z) + \lambda_{tk} p_{tk}(z) + \lambda_{tu} p_{tu}(z)} \, dz - \log \lambda_{tk} + \log 2 \\
&= \int_z p_{tk}(z) \log \frac{\lambda_{tk} p_{tk}(z)}{2p_{avg}(z)} \, dz - \log \lambda_{tk} + \log 2
\end{aligned}
$$

By multiplying $\lambda_{tk}$,

$$\lambda_{tk} D_{KL}(p_{tk} \| p_{avg}) = \int_z \lambda_{tk} p_{tk}(z) \log \frac{\lambda_{tk} p_{tk}(z)}{2p_{avg}(z)} \, dz - \lambda_{tk} \log \frac{\lambda_{tk}}{2}$$

Thus,

$$\int_z \lambda_{tk} p_{tk}(z) \log \frac{\lambda_{tk} p_{tk}(z)}{2p_{avg}(z)} \, dz = \lambda_{tk} D_{KL}(p_{tk} \| p_{avg}) + \lambda_{tk} \log \frac{\lambda_{tk}}{2} \tag{6}$$

Similarly, for the third term,

$$\int_z \lambda_{tu} p_{tu}(z) \log \frac{\lambda_{tu} p_{tu}(z)}{2p_{avg}(z)} \, dz = \lambda_{tu} D_{KL}(p_{tu} \| p_{avg}) + \lambda_{tu} \log \frac{\lambda_{tu}}{2} \tag{7}$$

In summary, from the Eq. (5), (6), and (7), we obtain the minimization problem with respect to $G$ as follows,

$$
\begin{aligned}
\min_{\theta_g} -\mathcal{L}_G(\theta_g, \theta_d) &= -\mathcal{L}_d^s(\theta_g, \theta_d) - \mathcal{L}_d^{tk}(\theta_g, \theta_d) + \mathcal{L}_d^{tu}(\theta_g, \theta_d) \\
&= D_{KL}(p_s \| p_{avg}) + \lambda_{tk} D_{KL}(p_{tk} \| p_{avg}) - \lambda_{tu} D_{KL}(p_{tu} \| p_{avg}) + C_0,
\end{aligned}
\tag{8}
$$

where $C_0 = -\log 2 + \lambda_{tk} \log \frac{\lambda_{tk}}{2} - \lambda_{tu} \log \frac{\lambda_{tu}}{2} = -2\lambda_{tk} \log 2 + \lambda_{tk} \log \lambda_{tk} - \lambda_{tu} \log \lambda_{tu}$. □

### B.1.4 Proof of Proposition 3.2

**Proposition B.2.** *The third term of the right-hand side in Eq. (14) in the main paper, $D_{KL}(p_{tu} \| p_{avg})$, is bounded to $\log 2 - \log \lambda_{tu}$ .*

*Proof.*

$$D_{KL}(p_{tu}\|p_{avg}) = \int_z p_{tu}(z) \log \frac{p_{tu}(z)}{p_{avg}(z)} \, dz$$

$$= \int_z p_{tu}(z) \log \frac{2p_{tu}(z)}{p_s(z) + \lambda_{tk}p_{tk}(z) + \lambda_{tu}p_{tu}(z)} \, dz$$

$$= \int_z p_{tu}(z) \log 2 \, dz + \int_z p_{tu}(z) \log \frac{p_{tu}(z)}{p_s(z) + \lambda_{tk}p_{tk}(z) + \lambda_{tu}p_{tu}(z)} \, dz$$

$$= \log 2 + \int_z p_{tu}(z) \log p_{tu}(z) \, dz - \int_z p_{tu}(z) \log \left( p_s(z) + \lambda_{tk}p_{tk}(z) + \lambda_{tu}p_{tu}(z) \right) dz$$

$$\leq \log 2 + \int_z p_{tu}(z) \log p_{tu}(z) \, dz - \int_z p_{tu}(z) \log \lambda_{tu}p_{tu}(z) \, dz$$

$$( \ p_s(z) + \lambda_k p_{tk}(z) + \lambda_{tu}p_{tu}(z) \geq \lambda_{tu}p_{tu}(z) \text{ for all } z )$$

$$= \log 2 + \int_z p_{tu}(z) \log p_{tu}(z) \, dz - \int_z p_{tu}(z) \log p_{tu}(z) \, dz - \int_z p_{tu}(z) \log \lambda_{tu} \, dz$$

$$= \log 2 - \log \lambda_{tu}$$

Therefore,

$$D_{KL}(p_{tu}\|p_{avg}) \leq \log 2 - \log \lambda_{tu}.$$

$\square$

### B.1.5 Proof of Proposition 3.3

**Proposition B.3.** *Assume that $supp(p_s) \cap supp(p_{tu}) = \emptyset$ and $supp(p_{tk}) \cap supp(p_{tu}) = \emptyset$, where $supp(p) := \{z \in \mathcal{Z} | p(z) > 0\}$ is the support set of probability distribution $p$. Then, the minimization problem with respect to $G$, Eq. (14) in the main paper, is equivalent to the minimization problem of summation on two $f$-divergences.*

$$D_{f_1}(p_s\|p_{tk}) + \lambda_{tk}D_{f_2}(p_{tk}\|p_s),$$

*where $f_1(u) = u \log \frac{u}{(1-\alpha)u+\alpha}$, and $f_2(u) = u \log \frac{u}{\alpha u + (1-\alpha)}$. Therefore, the minimum of Eq. (14) in the main paper is achieved if and only if $p_s = p_{tk}$.*

*Proof.* The minimization problem with respect to $G$ can be expressed as below:

$$D_{KL}(p_s\|p_{avg}) + \lambda_{tk}D_{KL}(p_{tk}\|p_{avg}) - \lambda_{tu}D_{KL}(p_{tu}\|p_{avg}) + C_0 \tag{9}$$

where $C_0 = -2\lambda_{tk} \log 2 + \lambda_{tk} \log \lambda_{tk} - \lambda_{tu} \log \lambda_{tu}$.

We assume that (i) $supp(p_s) \cap supp(p_{tu}) = \emptyset$ and (ii) $supp(p_{tk}) \cap supp(p_{tu}) = \emptyset$, where $supp(p) := \{z \in \mathcal{Z}|p(z) > 0\}$ be the support set of probability distribution $p$. We denote $\mathcal{Z}_1 := \mathcal{Z} \setminus supp(p_{tu})$ and $\mathcal{Z}_2 := supp(p_{tu})$. Then, the first and second term in Eq. (9) are written as below, respectively:

$$D_{KL}(p_s\|p_{avg}) = \int_{\mathcal{Z}} p_s(z) \log \frac{p_s(z)}{p_{avg}(z)} \, dz = \int_{\mathcal{Z}_1} p_s(z) \log \frac{p_s(z)}{(p_s(z) + \lambda_{tk}p_{tk}(z))/2} \, dz, \tag{10}$$

$$\lambda_{tk}D_{KL}(p_{tk}\|p_{avg}) = \lambda_{tk} \int_{\mathcal{Z}} p_{tk}(z) \log \frac{p_{tk}(z)}{p_{avg}(z)} \, dz \tag{11}$$

$$= \lambda_{tk} \int_{\mathcal{Z}_1} p_{tk}(z) \log \frac{p_{tk}(z)}{(p_s(z) + \lambda_{tk}p_{tk}(z))/2} \, dz. \tag{12}$$

since $p_{tu}(z) = 0$ for all $z \in \mathcal{Z}_1$ and $p_s(z) = p_{tk}(z) = 0$ for all $z \in \mathcal{Z}_2$. Also, the third term in Eq. (9) is as follows:

$$\lambda_{tu}D_{KL}(p_{tu}\|p_{avg}) = \lambda_{tu} \int_{\mathcal{Z}} p_{tu}(z) \log \frac{p_{tu}(z)}{p_{avg}(z)} \, dz$$

$$= \lambda_{tu} \int_{\mathcal{Z}_2} p_{tu}(z) \log \frac{p_{tu}(z)}{(\lambda_{tu}p_{tu}(z))/2} \, dz = \lambda_{tu} \log \frac{2}{\lambda_{tu}}, \tag{13}$$

With Eq. (10) to (11) and letting $C_2 := \lambda_{tu} \cdot \log \frac{2}{\lambda_{tu}}$ in Eq. (13), Eq. (9) is as below:

$$\int_{\mathcal{Z}_1} p_s(z) \log \frac{p_s(z)}{(p_s(z) + \lambda_{tk}p_{tk}(z))/2} \, dz + \lambda_{tk} \int_{\mathcal{Z}_1} p_{tk}(z) \log \frac{p_{tk}(z)}{(p_{tk}(z) + \lambda_{tk}p_{tk}(z))/2} \, dz - C_2$$

$$= \int_{\mathcal{Z}_1} p_s(z) \log \left[ \frac{p_s(z)}{(p_s(z) + \lambda_{tk}p_{tk}(z))/(1 + \lambda_{tk})} \cdot \frac{2}{1 + \lambda_{tk}} \right] dz$$

$$+ \lambda_{tk} \int_{\mathcal{Z}_1} p_{tk}(z) \log \left[ \frac{p_{tk}(z)}{(p_{tk}(z) + \lambda_{tk}p_{tk}(z))/(1 + \lambda_{tk})} \cdot \frac{2}{1 + \lambda_{tk}} \right] dz - C_2$$

$$= \int_{\mathcal{Z}_1} p_s(z) \log \frac{p_s(z)}{(p_s(z) + \lambda_{tk}p_{tk}(z))/(1 + \lambda_{tk})} \, dz + \int_{\mathcal{Z}_1} p_s(z) \log \frac{2}{1 + \lambda_{tk}} \, dz$$

$$+ \lambda_{tk} \int_{\mathcal{Z}_1} p_{tk}(z) \log \frac{p_{tk}(z)}{(p_{tk}(z) + \lambda_{tk}p_{tk}(z))/(1 + \lambda_{tk})} \, dz + \lambda_{tk} \int_{\mathcal{Z}_1} p_{tk}(z) \log \frac{2}{1 + \lambda_{tk}} \, dz - C_2$$

$$= \int_{\mathcal{Z}_1} p_s(z) \log \frac{p_s(z)}{(p_s(z) + \lambda_{tk}p_{tk}(z))/(1 + \lambda_{tk})} \, dz + \log \frac{2}{1 + \lambda_{tk}}$$

$$+ \lambda_{tk} \int_{\mathcal{Z}_1} p_{tk}(z) \log \frac{p_{tk}(z)}{(p_{tk}(z) + \lambda_{tk}p_{tk}(z))/(1 + \lambda_{tk})} \, dz + \lambda_{tk} \log \frac{2}{1 + \lambda_{tk}} - C_2$$

With denoting $\alpha := \frac{\lambda_{tk}}{1 + \lambda_{tk}}$, and $C_3 := \log \frac{2}{1+\lambda_{tk}} + \lambda_{tk} \log \frac{2}{1+\lambda_{tk}} - C_2$, and satisfying $0 < \alpha < 1$,

$$= \int_{\mathcal{Z}_1} p_s(z) \log \frac{p_s(z)}{(1 - \alpha)p_s(z) + \alpha p_{tk}(z)} \, dz + \lambda_{tk} \int_{\mathcal{Z}_1} p_{tk}(z) \log \frac{p_{tk}(z)}{(1 - \alpha)p_{tk}(z) + \alpha p_{tk}(z)} \, dz + C_3. \tag{14}$$

By the definition of the skewed $\alpha$-KL Divergence ($D_{KL}^{(\alpha)}$) [47], Eq. (14) is written as follow:

$$D_{KL}^{(\alpha)}(p_s \| p_{tk}) + \lambda_{tk} \cdot D_{KL}^{(1-\alpha)}(p_{tk} \| p_s) + C_3. \tag{15}$$

The skewed $\alpha$-KL Divergence, $D_{KL}^{(\alpha)}(p \| q)$, belongs to the $f$-divergence from $p$ to $q$ [47].

$$D_f(p \| q) = \int q(x) f\left(\frac{p(x)}{q(x)}\right) dx, \text{ where } f(u) = u \log \frac{u}{(1 - \alpha)u + \alpha}, \ (u = \frac{p(x)}{q(x)} \neq 1),$$

where $f(u)$ is a convex function with $f(1) = 0$. Therefore, Eq. (15) is equivalent to the summation of $f$-divergence as below.

$$D_{f_1}(p_s \| p_{tk}) + \lambda_{tk} D_{f_2}(p_{tk} \| p_s) + C_3, \tag{16}$$

where $f_1(u) = u \log \frac{u}{(1-\alpha)u+\alpha}$, and $f_2(u) = u \log \frac{u}{\alpha u+(1-\alpha)}$. Therefore, the minimum of Eq. (16) is achieved when $p_s = p_{tk}$. $\qquad \square$

## B.2 Posterior Inference

We provide the details of the posterior inference to estimate $w_x = p(known|x)$ for a target instance, $x$. Thus, we model the mixture of two Beta distributions on the entropy values of the target instances. We estimate $w_x$ as the posterior probability by fitting the Beta mixture model through the Expectation-Maximization (EM) algorithm. Therefore, this section starts the details of the fitting process of the Beta mixture model.

### B.2.1 Fitting Process of Beta mixture model

We follow the fitting process of Beta mixture model by [1]. First, the probability density function (pdf) for the mixture of two Beta distributions on the entropy values is defined as follows,

$$p(\ell_x) = \lambda_{tk} p(\ell_x | known) + \lambda_{tu} p(\ell_x | unknown), \tag{17}$$

$$\text{with} \quad p(\ell_x | known) \sim Beta(\alpha_0, \beta_0) \quad \text{and} \quad p(\ell_x | unknown) \sim Beta(\alpha_1, \beta_1), \tag{18}$$

where $\lambda_{tk}$ is $p(known)$; $\lambda_{tu}$ is $p(unknown)$; $\ell_x$ is the entropy value for the target instance, $x$, i.e. $\ell_x = H(E(G(x)))$ with entropy function $H$; $\alpha_0$ and $\beta_0$ represents the parameters of the Beta distribution for the *known* component; and $\alpha_1$ and $\beta_1$ are the parameters for *unknown* component. Eq. (18) represents the individual pdf for each component which is followed by the Beta distribution,

We fit the distribution through the Expectation-Maximization (EM) algorithm. We introduce the latent variables $\gamma_0(\ell_x) = p(known|\ell_x)$ and $\gamma_1(\ell_x) = p(unknown|\ell_x)$, and use an Expectation Maximization (EM) algorithm with a finite number of iterations (10 in ours).

In E-step, we update the latent variables using Bayes' rule with fixing the other parameters, $\lambda_{tk}$, $\alpha_0$, $\beta_0$, $\lambda_{tu}$, $\alpha_1$, and $\beta_1$, as follows:

$$\gamma_0(\ell_x) = \frac{\lambda_{tk}p(\ell_x|known)}{\lambda_{tk}p(\ell_x|known) + \lambda_{tu}p(\ell_x|unknown)},$$

where $p(\ell_x|known)$ and $p(\ell_x|unknown)$ are from Eq. (18). $\gamma_1(\ell_x)$ follows the same claculation.

In M-step, given the fixed $\gamma_0(\ell_x)$ and $\gamma_1(\ell_x)$ from the E-step, the parameters $\alpha_k$, $\beta_k$ are estimated by using a weighted method of moments as follows,

$$\beta_k = \frac{\alpha_k(1 - \bar{\ell}_k)}{\bar{\ell}_k}, \quad \alpha_k = \bar{\ell}_k(\frac{\bar{\ell}_k(1 - \bar{\ell}_k)}{s_k^2} - 1), \quad \text{where } k \in \{0, 1\},$$

where $\bar{\ell}_0$ and $s_0^2$ are a weighted average and a weighted variance estimation of the entropy values, $\ell_x$, for *known* component, respectively. $\bar{\ell}_1$ and $s_1^2$ are for *unknown* component as follows,

$$\bar{\ell}_k = \frac{\sum_{x \in \chi_t} \gamma_k(\ell_x)\ell_x}{\sum_{x \in \chi_t} \gamma_k(\ell_x)}, \quad s_k^2 = \frac{\sum_{x \in \chi_t} \gamma_k(\ell_x)(\ell_x - \bar{\ell}_k)^2}{\sum_{x \in \chi_t} \gamma_k(\ell_x)} \text{ where } k \in \{0, 1\}.$$

Then, the mixing coefficients, $\lambda_{tk}$ and $\lambda_{tu}$, are calculated as follows,

$$\lambda_{tk} = \frac{1}{n_t}\sum_{x \in \chi_t} \gamma_0(\ell_x), \quad \lambda_{tu} = 1 - \lambda_{tk}, \tag{19}$$

where $n_t$ is the number of instances in the target domain, $\chi_t$.

We conduct a finite number of iteration over E-step and M-step iteratively. Finally, the probability of a instance being *known* or *unknown* class through the posterior probability:

$$p(known|\ell_x) = \frac{\lambda_{tk}p(\ell_x|known)}{\lambda_{tk}p(\ell_x|known) + \lambda_{tu}p(\ell_x|unknown)}, \tag{20}$$

where $p(unknown|\ell_x)$ follows the same calculation.

## B.3 Training Details

This subsection provides the details for the training part of UADAL. The first part enumerates the training algorithm procedure, and the second part shows the computational complexity of UADAL during training.

### B.3.1 Training Algorithm of UADAL

We provide a training algorithm of UADAL in detail. All equations in Algorithm 1 of this script represent the equations in the main paper. The detailed settings for $n_{iter}$, $m$, and $\eta$ in Algorithm 1 are described in Section C.1.3.

### B.3.2 Computational Complexity of UADAL

The computational complexity of UADAL is increased because we need to fit the mixture model, which requires $O(nk)$, with the number of the target instances ($n$) and the iterations of EM ($k$). Figure 1 shows the negative log-likelihood from the fitted mixture model over EM iterations ($k$), along different initializations of $\lambda_{tk}$ and $\lambda_{tu}$. Here, $k$ can be adjusted to trade the performance (negative log-likelihood) and the time-complexity ($k$). Moreover, it shows that the convergence

**Algorithm 1** Training algorithm of UADAL.

---

**Require:**
    $\chi_s$: dataset from the source domain.
    $\chi_t$: dataset from the target domain.
    $n_{iter}$: the number of epochs for main training.
    $m$: the batch size.
    $\eta$: the frequency to fit the posterior inference.
**Ensure:**
  1: Sample few source minibatches from $\chi_s$.
  2: Update $\theta_e$, $\theta_g$ following Eq. (20).
  3: Fitting the posterior model through EM algorithm by Eq. (17)
  4: **for** $i = 1, \ldots, n_{iter}$ **do**
  5:     Sample minibatch of $m$ source samples from $\chi_s$.
  6:     Sample minibatch of $m$ target samples from $\chi_t$.
  7:     Update $\theta_d$ by Eq. (21).
  8:     Update $\theta_e$, $\theta_g$, $\theta_c$ by Eq. (20, 22).
  9:     **if** (i mod $\eta$) = 0 **then**
10:         Fitting the posterior model through EM algorithm by Eq. (17)
11:     **end if**
12: **end for**

---

of EM algorithm of the posterior inference, which makes we set $k$ as constant. As a measure of the computational complexity, we provide **Wall-clock-time** for a whole experimental procedure by following Algorithm 1 (under an RTX3090 GPU/i7-10700F CPU). For A→W in Office-31, the wall-clock-time is 1,484 and 1,415 seconds, with and without posterior inference, respectively (+5% increment). From this +5% increment, the performance of UADAL with the posterior inference has improved than with Entropy, as shown in Figure 9b in the main paper (see the red/blue solid lines).

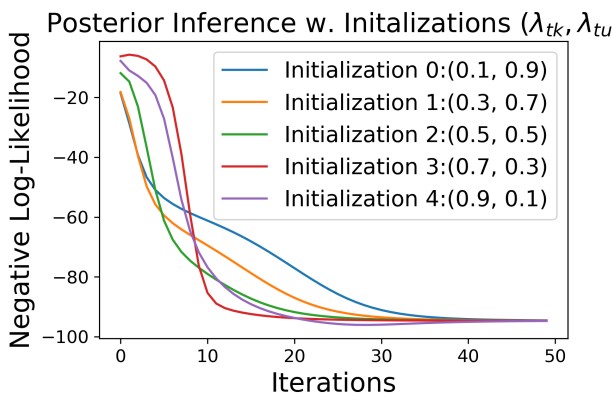

Figure 1: Convergence over EM iterations ($k$) of Posterior Inference

## C Experimental Part

### C.1 Implemenation details

#### C.1.1 Optimization Details

We utilize the pre-trained ResNet-50 [15], DenseNet-121 [16], EfficientNet-B0 [40], and VGGNet [37], as a backbone network. For all cases of the experiments for the backbone networks and the datasets, we use the SGD optimizer with the cosine annealing [26] schedule for the learning rate scheduling. For the parameters in the pre-trained network of ResNet-50, DenseNet, and EfficientNet, we set the learning rate 0.1 times smaller than the parameters from the scratch, followed by [20, 6]. For VGGNet with VisDA dataset, we followed [33]. Therefore, we did not update the parameters of VGGNet and constructed fully-connected layers with 100 hidden units after the FC8 layers. In terms of the entropy loss for the target domain, we adopt a variant of the loss, FixMatch [38], in order to utilize the confident predictions of the target instances. We run each setting **three times** and report the averaged accuracy with standard deviation. We conduct all experiments on an NVIDIA RTX 3090 GPU and an i7-10700F CPU.

#### C.1.2 Network Configurations

Except for the feature extractor network $G$, the configurations of the other networks, $E$, $C$, $D$, are based on the classification network. The feature dimensions from the ResNet-50, EfficientNet-B0, and DenseNet-121 are 2048, 1280, and 1024, respectively, which are the output dimensions of $G$ (100 for VGGNet by the construction of the fully connected layer). Given the dimension, the network $C$ utilizes one middle layer between the feature and classification layer with the dimension of 256. After the middle layer, we apply a batch normalization [17] and LeakyReLU [45] with 0.2 parameter. Then, the feature after the middle layer passes to the classification layer, in which the output dimensions are $|\mathcal{C}_s| + 1$. The network $E$ has the classification layer without any middle layer, in which the output dimension is $|\mathcal{C}_s|$. The network $D$ consists of the two middle layers with applying the LeakyReLU, where the output dimension is 3.

#### C.1.3 Hyperparameter Settings

For all cases of the experiments, we set the batch size, $m$, to 32. For the number of epochs for main training, denoted as $n_{iter}$, we set 100 for Office-31 and OfficeHome datasets. For the VisDA dataset, we set $n_{iter}$ as 10 epochs since it is a large scale. Associated with $n_{iter}$, we set the frequency to fit the posterior inference, $\eta$, as 10 epochs for Office-31 and OfficeHome, as 1 epoch for the VisDA dataset. We set 0.001 as a default learning rate with 0.1 times smaller for the network $G$ since we bring the pre-trained network. For the network $E$ of the open-set recognizer, we set 2 times larger than the default value because the shallow network $E$ should learn the labeled source domain quickly during $\eta$ epochs, followed by initializing the network $E$ after fitting the posterior inference. For the case of VGGNet with the VisDA dataset, we utilize the default learning rate for the network $G$ as same as the other networks since we only train the fully connected layers on the top of VGGNet.

#### C.1.4 Baselines

For the baselines, we implement the released codes for OSBP [33] (`https://github.com/ksaito-ut/OPDA_BP`), STA [20] (`https://github.com/thuml/Separate_to_Adapt`), PGL [27] (`https://github.com/BUserName/PGL`), ROS [6] (`https://github.com/silvia1993/ROS`), DANCE [34] (`https://github.com/VisionLearningGroup/DANCE`), and DCC [19] (`https://github.com/Solacex/Domain-Consensus-Clustering`). For OSLPP [44], we are not able to find the released code. Thus, the reported performances of OSLPP for Office-31 and Office-Home with ResNet-50 are only available. For the released codes, we follow their initial experimental settings. Especially, we set all experimental settings for DenseNet and EfficientNet, equal to the settings on their ResNet-50 experiments, since they do not conduct the experiments on the DenseNet-121 and EfficientNet-B0. For a **fair comparison**, we bring the reported results for the baselines from its papers on the datasets, i.e., Office-31 (with ResNet-50), OfficeHome (with ResNet-50), and VisDA(with VGGNet) dataset. The officially reported performances are marked as $^*$ in the tables. Except for these cases, we all re-implement the experiments three times.

### C.1.5 Dataset

For the availability of the datasets, we utilize the following links; Office-31 (`https://www.cc.gatech.edu/~judy/domainadapt/#datasets_code`), Office-Home (`https://www.hemanthdv.org/officeHomeDataset.html`), and VisDA (`http://ai.bu.edu/visda-2017/#download`). We utilize the data transformations for training the proposed model, which are 1) resize, random horizontal flip, crop, and normalize by following [34], and 2) RandAugment [9] by following [32].

## C.2 Experimental Results

### C.2.1 Computational Complexity of OSLPP on VisDA dataset

As OSLPP [44] said, their complexity is $\mathcal{O}(T(2n^2 d_{PCA} + d^3_{PCA}))$, which is repeated for $T$ times. Here, $n$ is the number of samples with $n = n_s + n_t$, and $d_{PCA}$ is the dimension which is reduced by PCA. However, regarding memory usage, when the number of samples, $n$, is much greater than the dimensionality, the memory complexity is $\mathcal{O}(n^2)$. Therefore, they claimed that it has limitation of scaling up to the extremely large dataset (e.g., $n > 100,000$). With this point, VisDA dataset consists of the source dataset with 79,765 instances and the target dataset with 55,388 instances, where the number of samples becomes 135,153. Therefore, OSLPP is infeasible to conduct the experiments for VisDA dataset.

### C.2.2 Low Accuracy of Baselines

HOS score is a harmonic mean of OS* and UNK. Therefore, HOS is higher when performing well in *both* known and unknown classification. With this point, some baselines have very low HOS score in the Table 1 and 2 of the main paper. This is because their UNK performances are worse. For example, the reported OS and OS* of PGL [27] in Office-Home are 74.0 and 76.1, respectively. Here, OS is the class-wise averaged accuracy over the classes including *unknown* class. With 25 known classes, UNK then becomes $(25+1) \times OS - 25 \times OS^* = 25.1$, which leads to HOS score as 33.5. It means that PGL fails in the open-set scenario because their adversarial loss includes *all* target instances, which is critically weakened by the negative transfer.

For DANCE, they only reported OS scores in the paper, which makes the calculation of HOS infeasible. Also, their class set (15 knowns in OfficeHome) is different from the standard OSDA scenario (25 knowns in OfficeHome by following [33]). It means that optimal hyper-parameters are not available. Therefore, we re-implement based on their official code. Meanwhile, DANCE (also DCC) is based on clustering which means that they are weak on initializations or hyperparameters, empirically shown as higher standard deviations of the performances in the tables. The below is the detailed answer on the lower performances of baselines, especially DANCE, with EfficientNet. As we said, there is no reported performance of DANCE with the additional backbone choices. Therefore, we implemented additional variants of DANCE with EfficientNet and DenseNet by following their officially released codes. In order to compare fairly, we set all hyper-parameters with that of ResNet-50 case as UADAL is being set. Specifically, DANCE requires a threshold value ($\rho$) to decide whether a target instance belongs to "known" class or not, which is very sensitive to the performance. We confirmed that they utilize the different values over the experimental settings. This sensitivity may degrade the performance of DANCE. Unlike DANCE, UADAL does not require a threshold setting because it has a posterior inference to automatically find the threshold to decide open-set instances. Therefore, this becomes the key reason behind the performance difference.

For DCC, the experimental settings for VisDA are not available with the comments of "the clustering on VisDA is not very stable" in their official code repository. Threfore, our re-implementation of DCC with VisDA (with EfficientNet-B0, DenseNet-121, and ResNet50) was also unstable. For a fair comparison with DCC, please refer to the performances which is marked as * in the Table 1 and 2 of the main paper.

### C.2.3 t-SNE Visualization

Figure 2 and 3 in this script represents the t-SNE visualizations of the learned features extracted by EfficientNet-B0 and ResNet-50, respectively. It should be noted that EfficientNet-B0 (5.3M) has only 20% of parameters than ResNet-50 (25.5M). We observe that the target-unknown (red) features from

the baselines are not discriminated with the source (green) and the target-known (blue) features. On the contrary, UADAL and cUADAL align the features from the source and the target-known instances accurately with clear *segregation* of the target-unknown features. It means that UADAL learns the feature spaces effectively even in the less complexity.

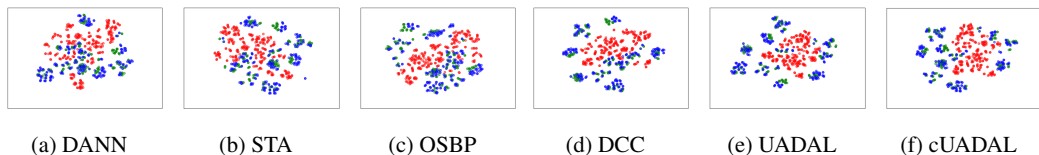

(a) DANN (b) STA (c) OSBP (d) DCC (e) UADAL (f) cUADAL

Figure 2: t-SNE of the features by the EfficientNet-B0 on the task D → W of Office-31. (blue: target-known, red: target-unkonwn, green: source)

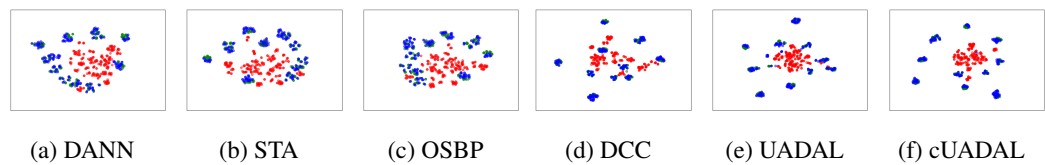

(a) DANN (b) STA (c) OSBP (d) DCC (e) UADAL (f) cUADAL

Figure 3: t-SNE of the features by the ResNet-50 on the task D → W of Office-31. (blue: target-known, red: target-unkonwn, green: source)

#### C.2.4 Proxy $\mathcal{A}$-Distance (PAD)

Proxy $\mathcal{A}$-Distance (PAD) is an empirical measure of distance between domain distributions, which is proposed by [11]. Given a generalization error $\epsilon$ of discriminating data which sampled by the domain distributions, PAD value can be computed as $\hat{d}_\mathcal{A} = 2(1 - 2\epsilon)$. We compute the PAD value between target-known and target-unknown features from the feature extractor, $G$. We follow the detailed procedure in [11]. Note that high PAD value means two domain distributions are well discriminated.

#### C.2.5 Robust on Early Stage Iterations

We investigate the effects of the learning iterations for the early stage training to fit the posterior inference, which is considered as a hyper-parameter of UADAL. Figure 4 represents the performance metrics such as OS*, UNK, and HOS over the number of the initial training iterations of UADAL. It shows that UADAL is not sensitive to the number of iterations for the early stage. Taken together with Figure 8a in the main paper, these results represent that our two modality assumption for the target entropy values robustly holds.

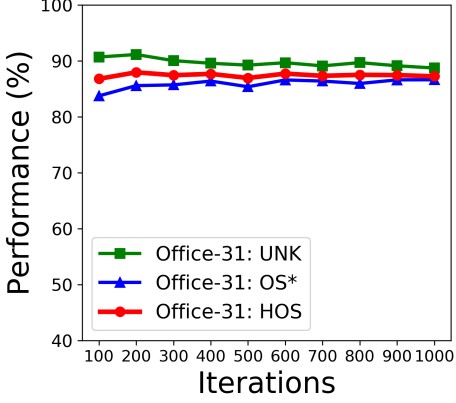

Figure 4: Averaged performance over the tasks in Office-31 varying the number of iterations.

| | A → W | | | A → D | | | D → W | | | W → D | | | D → A | | | W → A | | | Avg. | | |
|---|---|---|---|---|---|---|---|---|---|---|---|---|---|---|---|---|---|---|---|---|---|
| Network | OS* | UNK | HOS | OS* | UNK | HOS | OS* | UNK | HOS | OS* | UNK | HOS | OS* | UNK | HOS | OS* | UNK | HOS | OS* | UNK | HOS |
| C | 80.5 | 92.4 | 86.1 | 79.4 | 91.0 | 84.8 | 99.0 | 98.3 | **98.6** | 98.7 | 100.0 | 99.3 | 68.7 | 89.7 | 77.9 | 56.5 | 89.1 | 68.9 | 80.5 | 93.4 | 85.9 |
| E | 84.3 | 94.5 | **89.1** | 85.1 | 87.0 | **86.0** | 99.3 | 96.3 | 97.8 | 99.5 | 99.4 | **99.5** | 73.3 | 87.3 | **79.7** | 67.4 | 88.4 | **76.5** | 84.8 | 92.1 | **88.1** |

**Office-Home (ResNet-50)**

| | P→R | | | P→C | | | P→A | | | A→P | | | A→R | | | A→C | | |
|---|---|---|---|---|---|---|---|---|---|---|---|---|---|---|---|---|---|---|
| Network | OS* | UNK | HOS | OS* | UNK | HOS | OS* | UNK | HOS | OS* | UNK | HOS | OS* | UNK | HOS | OS* | UNK | HOS |
| C | 68.8 | 83.7 | 75.5 | 39.3 | 79.4 | 52.6 | 47.9 | 84.3 | 61.1 | 65.0 | 76.3 | 70.2 | 78.4 | 75.6 | 77.0 | 46.7 | 77.2 | 58.2 |
| E | 71.6 | 83.1 | **76.9** | 43.4 | 81.5 | **56.6** | 50.5 | 83.7 | **63.0** | 69.1 | 72.5 | **70.8** | 81.3 | 73.7 | **77.4** | 54.9 | 74.7 | **63.2** |

| | R→A | | | R→P | | | R→C | | | C→R | | | C→A | | | C→P | | | Avg. | | |
|---|---|---|---|---|---|---|---|---|---|---|---|---|---|---|---|---|---|---|---|---|---|
| Network | OS* | UNK | HOS | OS* | UNK | HOS | OS* | UNK | HOS | OS* | UNK | HOS | OS* | UNK | HOS | OS* | UNK | HOS | OS* | UNK | HOS |
| C | 64.2 | 79.2 | 71.0 | 75.5 | 78.6 | **77.0** | 44.9 | 71.7 | 55.2 | 63.0 | 75.0 | 68.5 | 49.9 | 77.4 | 60.7 | 57.8 | 79.2 | 66.8 | 58.4 | 78.1 | 66.1 |
| E | 66.7 | 78.6 | **72.1** | 77.4 | 76.2 | 76.8 | 51.1 | 74.5 | **60.6** | 69.1 | 78.3 | **73.4** | 53.5 | 80.5 | **64.2** | 62.1 | 78.8 | **69.5** | 62.6 | 78.0 | **68.7** |

Table 1: Ablation study for the network $C$ and $E$ to generate the entropy value in UADAL w.r.t. classification accuracies (%) on Office-31 and Office-Home wiht ResNet-50 (bold: best performer).

### C.2.6   Ablation Studies on Using Classifier $C$ as Entropy

This part introduces an ablation study for utilizing the classifier $C$ to generate the entropy values instead of an open-set recognizer $E$. Using the classifier $C$ (except the last unknown dimension) directly is also feasible, as $E$ does. Although it is feasible, however, it leads to wrong decisions for the target-unknown instances. As we explained, the network $E$ learns the decision boundary over $C_s$ classes while the network $C$ does over $C_s + 1$ including *unknown* class. Especially, in the case of the target-unknown instances, the network $C$ is enforced to classify the instances as *unknown class*, which is $C_s + 1$-th dimension. When optimizing the network $C$ with the target-unknown instances, we expect that the output of the network $C$ would have a higher value on the unknown dimension. With this point, if we use the first $C_s$ dimension of $C$ to calculate the entropy value, there is no evidence that the distribution of the $C$'s output is flat over $C_s$ classes which implies to a higher entropy value. Even though the largest predicted value except the last dimension is small, the entropy value might be lower due to imbalance in the output. Then, it becomes to be considered as *known* class, which is wrong decision for the target-unknown instances. Therefore, it gives the negative effects on the open-set recognition, and it adversely affects when training the model.

As an ablation experiment, we conduct the experiments to compare using $E$ or $C$ for the entropy. The experimental results on both cases are shown in Table 1 of this section, applied to Office-31 and Offce-Home datasets. The network $E$ in Table 1 is the current UADAL model and $C$ represents that the entropy values are generated by the classifier $C$ without introducing the network $E$. As you can see, the performances with $E$ is better than $C$. It means that the network $E$ learns the decision boundary for the known classes, and it leads to recognize the open-set instances effectively. It should be noted that we utilize the structure of $E$ as an one-layered network to reduce the computation burden.

### C.2.7   Ablation Studies on Entropy Minimization

The entropy minimization is important part for the fields such as semi-supervised learning [14, 38] and domain adaptation [24, 20, 34, 6] where the label information of the dataset is not available. In order to show the effect of this term, we conduct the ablation study on the datasets of Office-31 and Office-Home. We provide the experimental results in Table 2. Combined with the results in the main paper, the experimental result shows that UADAL without the entropy minimization loss still performs better than other baselines. It represents that UADAL learns the feature space appropriately as we intended to suit Open-Set Domain Adaptation. The properly learned feature space leads to effectively classify the target instances without the entropy minimization.

### C.2.8   Posterior Inference with Efficiency

In terms of complexity, the posterior inference increases the computational complexity because we need to fit the mixture model. As we provided at the section B.3.2, Wall-clock-time is increased as 5% with the posterior inference in the case of full data utilization. From this +5% increment, the performance has improved significantly than that without the posterior inference (as shown in Figure 9b in the paper). In addition, by utilizing the posterior inference, we avoid introducing any extra hyper-parameter to recognize the unknown instances, which is also our contribution.

As an alternative, we fit the mixture model only by sampling the target instances in order to reduce the computation time because the computational complexity is $O(nk)$ where $n$ is the number of samples

Table 2 (Office31 part):

| Entropy Minimization | A → W | | | A → D | | | D → W | | | W → D | | | D → A | | | W → A | | | Avg. | | |
|---|---|---|---|---|---|---|---|---|---|---|---|---|---|---|---|---|---|---|---|---|---|
| | OS* | UNK | HOS | OS* | UNK | HOS | OS* | UNK | HOS | OS* | UNK | HOS | OS* | UNK | HOS | OS* | UNK | HOS | OS* | UNK | HOS |
| X | 85.9 | 84.4 | 85.1 | 84.7 | 83.6 | 84.2 | 95.6 | 98.9 | 97.2 | 98.7 | 100.0 | 99.3 | 75.2 | 86.0 | **80.3** | 72.6 | 87.4 | **79.3** | 85.4 | 90.0 | 87.5 |
| O | 84.3 | 94.5 | **89.1** | 85.1 | 87.0 | **86.0** | 99.3 | 96.3 | **97.8** | 99.5 | 99.4 | **99.5** | 73.3 | 87.3 | 79.7 | 67.4 | 88.4 | 76.5 | 84.8 | 92.1 | **88.1** |

Office-Home (ResNet-50)

| Entropy Minimization | P→R | | | P→C | | | P→A | | | A→P | | | A→R | | | A→C | | |
|---|---|---|---|---|---|---|---|---|---|---|---|---|---|---|---|---|---|---|
| | OS* | UNK | HOS | OS* | UNK | HOS | OS* | UNK | HOS | OS* | UNK | HOS | OS* | UNK | HOS | OS* | UNK | HOS |
| X | 70.7 | 78.2 | 74.2 | 48.4 | 76.6 | **59.3** | 49.9 | 76.2 | 60.3 | 64.5 | 79.4 | **71.2** | 78.8 | 75.2 | 77.0 | 56.3 | 75.1 | **64.3** |
| O | 71.6 | 83.1 | **76.9** | 43.4 | 81.5 | 56.6 | 50.5 | 83.7 | **63.0** | 69.1 | 72.5 | 70.8 | 81.3 | 73.7 | **77.4** | 54.9 | 74.7 | 63.2 |

| | R→A | | | R→P | | | R→C | | | C→R | | | C→A | | | C→P | | | Avg. | | |
|---|---|---|---|---|---|---|---|---|---|---|---|---|---|---|---|---|---|---|---|---|---|
| | OS* | UNK | HOS | OS* | UNK | HOS | OS* | UNK | HOS | OS* | UNK | HOS | OS* | UNK | HOS | OS* | UNK | HOS | OS* | UNK | HOS |
| X | 62.3 | 76.4 | 68.6 | 71.3 | 81.0 | 75.8 | 57.3 | 65.6 | **61.2** | 68.1 | 75.9 | 71.8 | 54.7 | 70.9 | 61.7 | 60.1 | 72.6 | 65.8 | 61.9 | 75.2 | 67.6 |
| O | 66.7 | 78.6 | **72.1** | 77.4 | 76.2 | **76.8** | 51.1 | 74.5 | 60.6 | 69.1 | 78.3 | **73.4** | 53.5 | 80.5 | **64.2** | 62.1 | 78.8 | **69.5** | 62.6 | 78.0 | **68.7** |

Table 2: Ablation study for the entropy minimization loss in UADAL w.r.t. classification accuracies (%) on Office-31 and Office-Home wiht ResNet-50 (bold: best performer).

and $k$ is the number of fitting iterations (we fixed it as 10). Figure 5 represents the wall-clock time and the performance measures by sampling ratio (%) for the target domain. Since the computational complexity is linearly increased by the number of samples, the wall-clock time is also linearly increased by increasing the sampling ratio. Interestingly, we observed that even though the sampling ratio is small, i.e. 10%, the performances of UADAL w.r.t. HOS, OS*, and UNK does not decreased, on both Office-31 and Office-Home datasets.

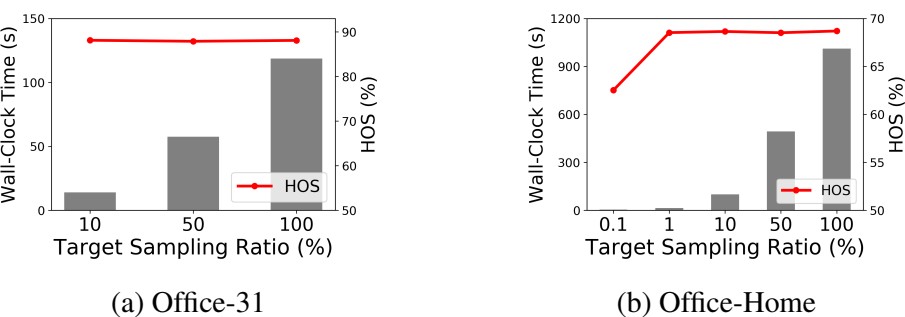

(a) Office-31          (b) Office-Home

Figure 5: Quantitative analysis for ablation study of applying sampling on the target domain with Office-31 (a) and Office-Home (b). Each subfigure represents the Wall-Clock Time (s) increased by fitting process of the mixture model during training and HOS (%) over the target sampling ratio. (All records are the averaged values over the tasks in each dataset, not just single task.)

In order to investigate the robustness on the sampling ratio, we provide the qualitative analysis in Figure 6. For each sampling ratio, the left figure represents the original target entropy distribution, and the middle shows the sampled target entropy values and the fitted BMM densities. Finally, the right figure represents the weight distribution by the posterior inference. As you can see, our posterior inference takes the entropy values, and fits the mixture model without any thresholds. Therefore, even if the sampling ratio is small, the observation that the target-unknown instances have higher entropy values than the target-known instances still holds. Therefore, the open-set recognition on the target domain is still informative, and it leads to maintain the performances of UADAL.

### C.2.9 Full Experimental Results with All Metrics

As a reminder, HOS metric is a harmonic mean of OS* and UNK where OS* is accuracy for the known class classification and UNK is for the unknown classification. Since Open-Set Domain Adaptation should perform well on both tasks, we choose HOS metric as a primary metric. For other metrics such as OS, OS*, and UNK, we provide the full experimental results including OS, OS*, and UNK in this section. First of all, we provide the summary table of the experimental results with the officially reported performances of the baselines, which is denoted as * for reliable and fair comparisons. Table 3 in this appendix shows that UADAL outperforms the baselines over all datasets, in the conventional setting of the backbone networks (such as Office-31/Office-Home with ResNet-50 and VisDA with VGGNet). The detailed results are shown in Table 4 for Office-31 and Table 5 for Office-Home, in this appendix.

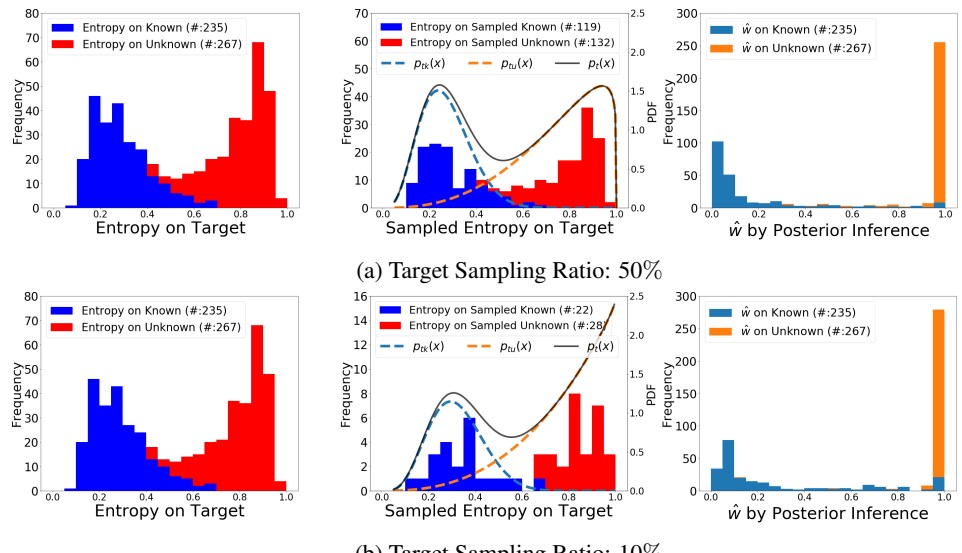

(a) Target Sampling Ratio: 50%

(b) Target Sampling Ratio: 10%

Figure 6: Qualitative analysis for ablation study of applying sampling on the target domain (D→W task in Office-31). The subfigure (a) and (b) represent the different sampling ratio, 50% and 10%, respectively. Each subfigure consists of 1) left: the original target entropy distribution, 2) midde: the sampled target entropy distribution with the fitted Beta Mixture Model (BMM), and 3) right: the weight ($\hat{w}$) distribution (by the fitted BMM in middle) on the target domain.

| Method | Office31 (ResNet-50) | | | | Office-Home (ResNet-50) | | | | VisDA (VGGNet) | | | |
|---|---|---|---|---|---|---|---|---|---|---|---|---|
| | OS | OS* | UNK | Avg. HOS | OS | OS* | UNK | Avg. HOS | OS | OS* | UNK | HOS |
| DANN | 85.4 | 87.1 | 68.3 | 75.9±0.5 | 53.5 | 52.6 | 77.1 | 60.7±0.2 | - | - | - | - |
| CDAN | 86.1 | 88.3 | 63.9 | 73.4±1.3 | 55.3 | 54.5 | 74.6 | 61.4±0.3 | - | - | - | - |
| OSBP* | 86.6 | 87.2 | 80.4 | 83.7±0.4 | 64.2 | 64.1 | 66.3 | 64.7±0.2 | 62.9 | 59.2 | 85.1 | 69.8 |
| STA* | 82.5 | 84.3 | 64.8 | 72.5±0.8 | 61.9 | 61.8 | 63.3 | 61.1±0.3 | 66.8 | 63.9 | 84.2 | 72.7 |
| PGL* | 81.1 | 82.7 | 64.7 | 72.6±1.5 | 74.1 | 76.1 | 25.0 | 35.2 | 80.7 | 82.8 | 68.1 | 74.7 |
| ROS* | 86.5 | 86.6 | 85.8 | 85.9±0.2 | 62.0 | 61.6 | 72.4 | 66.2±0.3 | - | - | - | - |
| DANCE | 91.0 | 94.0 | 60.2 | 73.1±1.0 | 72.8 | 74.4 | 35.0 | 44.2±0.6 | - | - | - | - |
| DCC* | - | - | - | 86.8 | - | - | - | 64.2 | 68.8 | 68.0 | 73.6 | 70.7 |
| LGU* | - | - | - | - | 71.4 | 72.7 | 38.9 | 50.7 | 70.1 | 69.2 | 75.5 | 72.2 |
| OSLPP* | 89.0 | 89.3 | 85.6 | 87.4 | 64.1 | 63.8 | 71.7 | 67.0 | - | - | - | - |
| **UADAL** | 85.5 | 84.8 | 92.1 | 88.1±0.2 | 63.1 | 62.6 | 78.0 | **68.7±0.2** | 67.4 | 63.1 | 93.3 | 75.3 |
| **cUADAL** | 85.6 | 84.8 | 93.0 | **88.5±0.3** | 63.1 | 62.5 | 77.6 | 68.5±0.1 | 68.3 | 64.3 | 92.6 | **75.9** |

Table 3: Summary of the OSDA experimental results. The results in Office-31 and Office-Home are the averaged accuracies over the tasks because there are the multiple domains. (bold: best performer, underline: second-best performer, *: officially reported performances.)

# D  Limitations and Potential Negative Societal Impacts

**Limitations** Our domain adaptation setting assumes that we have an access to a labeled source dataset and an unlabeled target dataset, simultaneously. Thus, we may encounter the situation where the access for the source dataset and the target dataset is not available at the same time, i.e. streamlined data gathering. In addition, our work solves the Open-Set Domain Adaptation problem. It intrinsically assumes the existence of 'unknown' information in the target domain. Our open-set recognition is based on this assumption, thus we fit the mixture model where each mode represents for known/unknown information. We think that the common assumption of the high entropy value on target-unknowns could be considered as a limitation, as well.

**Potential Negative Societal Impacts** Because open-set domain adaptation focuses on categories belonging to the class of the source dataset, it is infeasible to distinguish differences between categories that are only within the target dataset. Therefore, if the source dataset's categories are not sufficient, important categories within the target dataset may not be classified, which would lead to only limited applications when we have social stratifications.

**Office31 (ResNet-50)**

| Model | A→W OS | OS* | UNK | HOS | A→D OS | OS* | UNK | HOS | D→W OS | OS* | UNK | HOS |
|---|---|---|---|---|---|---|---|---|---|---|---|---|
| DANN | 84.5 | 87.4 | 55.7 | 68.1 | 87.9 | 90.8 | 59.2 | 71.5 | 97.3 | 99.3 | 77.0 | 86.7 |
| CDAN | 86.7 | 90.3 | 50.7 | 64.9 | 88.6 | 92.2 | 52.4 | 66.8 | 97.2 | 99.6 | 73.2 | 84.3 |
| OSBP | 86.1 | 86.8 | 79.2 | 82.7 | 89.1 | 90.5 | 75.5 | 82.4 | 97.6 | 97.7 | 96.7 | 97.2 |
| STA | 85.0 | 86.7 | 67.6 | 75.9 | 88.5 | 91.0 | 63.9 | 75.0 | 90.6 | 94.1 | 55.5 | 69.8 |
| PGL | 81.4 | 82.7 | 67.9 | 74.6 | 80.5 | 82.1 | 65.4 | 72.8 | 85.7 | 87.5 | 68.1 | 76.5 |
| ROS | 87.3 | 88.4 | 76.7 | 82.1 | 86.6 | 87.5 | 77.8 | 82.4 | 98.7 | 99.3 | 93.0 | 96.0 |
| DANCE | 94.3 | 98.7 | 50.7 | 66.9 | 92.8 | 96.5 | 55.9 | 70.7 | 97.0 | 100.0 | 66.8 | 80.0 |
| DCC | - | - | - | 87.1 | - | - | - | 85.5 | - | - | - | 91.2 |
| LGU | - | - | - | - | - | - | - | - | - | - | - | - |
| OSLPP | 89.4 | 89.5 | 88.4 | 89.0 | 92.4 | 92.6 | 90.4 | **91.5** | 96.1 | 96.9 | 88.0 | 92.3 |
| **UADAL** | 85.3 | 84.3 | 94.5 | 89.1 | 85.2 | 85.1 | 87.0 | 86.0 | 99.0 | 99.3 | 96.3 | 97.8 |
| **cUADAL** | 86.4 | 85.5 | 95.1 | 90.1 | 86.0 | 85.6 | 90.4 | 87.9 | 98.6 | 98.7 | 97.7 | 98.2 |

| Model | W→D OS | OS* | UNK | HOS | D→A OS | OS* | UNK | HOS | W→A OS | OS* | UNK | HOS | AVG. OS | OS* | UNK | HOS |
|---|---|---|---|---|---|---|---|---|---|---|---|---|---|---|---|---|
| DANN | 97.3 | 100.0 | 70.2 | 82.5 | 73.0 | 72.9 | 74.5 | 73.7 | 72.2 | 72.1 | 73.1 | 72.6 | 85.4 | 87.1 | 68.3 | 75.9±0.5 |
| CDAN | 97.0 | 100.0 | 67.3 | 80.5 | 74.5 | 74.9 | 70.6 | 72.7 | 72.5 | 72.8 | 69.3 | 71.0 | 86.1 | 88.3 | 63.9 | 73.4±1.3 |
| OSBP | 97.7 | 99.1 | 84.2 | 91.1 | 75.8 | 76.1 | 72.3 | 75.1 | 73.1 | 73.0 | 74.4 | 73.7 | 86.6 | 87.2 | 80.4 | 83.7±0.4 |
| STA | 83.3 | 84.9 | 67.8 | 75.2 | 81.5 | 83.1 | 65.9 | 73.2 | 66.4 | 66.2 | 68.0 | 66.1 | 82.5 | 84.3 | 64.8 | 72.5±0.8 |
| PGL | 81.1 | 82.8 | 64.0 | 72.2 | 78.8 | 80.6 | 61.2 | 69.5 | 79.1 | 80.8 | 61.8 | 70.1 | 81.1 | 82.7 | 64.7 | 72.6±1.5 |
| ROS | 99.9 | 100.0 | 99.4 | **99.7** | 75.4 | 74.8 | 81.2 | 77.9 | 71.2 | 69.7 | 86.6 | 77.2 | 86.5 | 86.6 | 85.8 | 85.9±0.5 |
| DANCE | 97.6 | 100.0 | 73.7 | 84.8 | 82.4 | 85.3 | 53.6 | 65.8 | 81.6 | 83.7 | 60.6 | 70.2 | 91.0 | 94.0 | 60.2 | 73.1±1.0 |
| DCC | - | - | - | 87.1 | - | - | - | **85.5** | - | - | - | **84.4** | - | - | - | 86.8 |
| LGU | - | - | - | - | - | - | - | - | - | - | - | - | - | - | - | - |
| OSLPP | 95.4 | 95.8 | 91.5 | 93.6 | 81.6 | 82.1 | 76.6 | 79.3 | 78.9 | 78.9 | 78.5 | 78.7 | 89.0 | 89.3 | 85.6 | 87.4 |
| **UADAL** | 99.5 | 99.5 | 99.4 | 99.5 | 74.5 | 73.3 | 87.3 | 79.7 | 69.3 | 67.4 | 88.4 | 76.5 | 85.5 | 84.8 | 92.1 | 88.1±0.2 |
| **cUADAL** | 99.3 | 99.3 | 99.4 | 99.4 | 75.4 | 74.2 | 87.8 | 80.5 | 67.6 | 65.6 | 87.8 | 75.1 | 85.6 | 84.8 | 93.0 | **88.5±0.3** |

Table 4: Classification accuracy (%) on Office-31 dataset using ResNet-50 as the backbone network. (bold: best performer, underline: second-best performer)

**Office-Home (ResNet-50)**

| Model | P→R OS | OS* | UNK | HOS | P→C OS | OS* | UNK | HOS | P→A OS | OS* | UNK | HOS | A→P OS | OS* | UNK | HOS |
|---|---|---|---|---|---|---|---|---|---|---|---|---|---|---|---|---|
| DANN | 67.9 | 67.7 | 72.0 | 69.8 | 32.3 | 30.1 | 86.3 | 44.6 | 44.0 | 42.4 | 83.9 | 56.3 | 60.4 | 60.0 | 71.3 | 65.2 |
| CDAN | 69.8 | 69.8 | 69.7 | 69.7 | 35.0 | 33.1 | 82.4 | 47.2 | 47.1 | 45.8 | 81.2 | 58.6 | 62.0 | 61.7 | 68.8 | 65.1 |
| OSBP | 76.0 | 76.2 | 71.7 | 73.9 | 45.3 | 44.5 | 66.3 | 53.2 | 59.4 | 59.1 | 68.1 | 63.2 | 71.3 | 71.8 | 59.8 | 65.2 |
| STA | 75.7 | 76.2 | 64.3 | 69.5 | 45.1 | 44.2 | 67.1 | 53.2 | 54.9 | 54.2 | 72.4 | 61.9 | 67.2 | 68.0 | 48.4 | 54.0 |
| PGL | 82.6 | 84.8 | 27.6 | 41.6 | 58.4 | 59.2 | 38.4 | 46.6 | 72.2 | 73.7 | 34.7 | 47.2 | 77.1 | 78.9 | 32.1 | 45.6 |
| ROS | 71.1 | 70.8 | 78.4 | 74.4 | 47.5 | 46.5 | 71.2 | 56.3 | 57.6 | 57.3 | 64.3 | 60.6 | 68.5 | 68.4 | 70.3 | 69.3 |
| DANCE | 84.2 | 86.5 | 27.1 | 41.2 | 48.9 | 48.2 | 67.4 | 55.7 | 69.7 | 70.7 | 43.9 | 54.2 | 82.2 | 84.0 | 35.4 | 49.8 |
| DCC | - | - | - | 64.0 | - | - | - | 52.8 | - | - | - | 59.5 | - | - | - | 67.4 |
| LGU | 81.2 | 82.8 | 41.2 | 55.0 | 53.1 | 54.5 | 18.1 | 27.2 | 68.4 | 69.1 | 50.9 | 58.6 | 79.3 | 80.5 | 49.3 | 61.2 |
| OSLPP | 76.8 | 77.0 | 71.2 | 74.0 | 53.6 | 53.1 | 67.1 | 59.3 | 55.4 | 54.6 | 76.2 | 63.6 | 72.5 | 72.5 | 73.1 | 72.8 |
| **UADAL** | 72.1 | 71.6 | 83.1 | 76.9 | 44.9 | 43.4 | 81.5 | 56.6 | 51.8 | 50.5 | 83.7 | 63.0 | 69.2 | 69.1 | 72.5 | 70.8 |
| **cUADAL** | 71.7 | 71.2 | 83.4 | 76.8 | 42.7 | 41.2 | 80.7 | 54.6 | 52.1 | 50.9 | 82.4 | 62.9 | 69.6 | 69.4 | 73.9 | 71.6 |

| Model | A→R OS | OS* | UNK | HOS | A→C OS | OS* | UNK | HOS | R→A OS | OS* | UNK | HOS | R→P OS | OS* | UNK | HOS |
|---|---|---|---|---|---|---|---|---|---|---|---|---|---|---|---|---|
| DANN | 74.8 | 75.1 | 67.3 | 71.0 | 38.9 | 37.1 | 82.7 | 51.2 | 57.6 | 56.8 | 77.1 | 65.4 | 69.5 | 69.6 | 67.2 | 68.4 |
| CDAN | 74.8 | 75.2 | 66.7 | 70.7 | 41.2 | 39.7 | 78.9 | 52.9 | 60.4 | 59.8 | 73.6 | 66.0 | 70.6 | 70.9 | 64.6 | 67.6 |
| OSBP | 78.8 | 79.3 | 67.5 | 72.9 | 50.6 | 50.2 | 61.1 | 55.1 | 66.1 | 66.1 | 67.3 | 66.7 | 76.0 | 76.3 | 68.6 | 72.3 |
| STA | 77.9 | 78.6 | 60.4 | 68.3 | 47.0 | 46.0 | 72.3 | 55.8 | 67.5 | 67.5 | 66.7 | 67.1 | 76.3 | 77.1 | 55.4 | 64.5 |
| PGL | 85.9 | 87.7 | 40.9 | 55.8 | 61.6 | 63.3 | 19.1 | 29.3 | 78.6 | 81.5 | 6.1 | 11.4 | 83.0 | 84.8 | 38.0 | 52.5 |
| ROS | 75.9 | 75.8 | 77.2 | 76.5 | 51.5 | 50.6 | 74.1 | 60.1 | 67.1 | 67.0 | 70.8 | 68.8 | 72.3 | 72.0 | 80.0 | 75.7 |
| DANCE | 87.4 | 89.8 | 25.3 | 39.4 | 54.4 | 54.4 | 53.7 | 53.1 | 76.8 | 79.2 | 16.7 | 27.5 | 84.1 | 86.2 | 29.6 | 44.0 |
| DCC | - | - | - | 80.6 | - | - | - | 52.9 | - | - | - | 56.0 | - | - | - | 62.7 |
| LGU | 85.0 | 86.5 | 47.5 | 61.3 | 57.6 | 58.6 | 32.6 | 41.9 | 76.4 | 77.5 | 48.9 | 60.0 | 81.8 | 83.2 | 46.8 | 59.9 |
| OSLPP | 79.7 | 80.1 | 69.4 | 74.3 | 56.3 | 55.9 | 67.1 | 61.0 | 61.3 | 60.8 | 75.0 | 67.2 | 78.1 | 78.4 | 70.8 | 74.4 |
| **UADAL** | 81.0 | 81.3 | 73.7 | 77.4 | 55.7 | 54.9 | 74.7 | 63.2 | 67.1 | 66.7 | 78.6 | 72.1 | 77.3 | 77.4 | 76.2 | 76.8 |
| **cUADAL** | 81.8 | 82.2 | 73.3 | 77.5 | 55.8 | 55.0 | 75.6 | 63.6 | 67.3 | 66.8 | 79.6 | 72.6 | 77.7 | 77.8 | 75.6 | 76.7 |

| Model | R→C OS | OS* | UNK | HOS | C→R OS | OS* | UNK | HOS | C→A OS | OS* | UNK | HOS | C→P OS | OS* | UNK | HOS | AVG. OS | OS* | UNK | HOS |
|---|---|---|---|---|---|---|---|---|---|---|---|---|---|---|---|---|---|---|---|---|
| DANN | 38.8 | 37.1 | 80.9 | 50.9 | 61.6 | 61.1 | 73.5 | 66.7 | 45.4 | 43.8 | 84.3 | 57.6 | 51.2 | 50.1 | 77.6 | 60.9 | 53.5 | 52.6 | 77.1 | 60.7±0.2 |
| CDAN | 41.7 | 40.3 | 75.8 | 52.7 | 62.0 | 61.5 | 73.7 | 67.1 | 46.4 | 44.9 | 82.8 | 58.2 | 52.6 | 51.6 | 76.8 | 61.7 | 55.3 | 54.5 | 74.6 | 61.4±0.3 |
| OSBP | 48.6 | 48.0 | 63.0 | 54.5 | 71.9 | 72.0 | 69.2 | 70.6 | 59.8 | 59.4 | 70.3 | 64.3 | 66.8 | 67.0 | 62.7 | 64.7 | 64.2 | 64.1 | 66.3 | 64.7±0.2 |
| STA | 50.3 | 49.9 | 61.1 | 54.5 | 67.0 | 67.0 | 66.7 | 66.8 | 51.9 | 51.4 | 65.0 | 57.4 | 61.7 | 61.8 | 59.1 | 60.4 | 61.9 | 61.8 | 63.3 | 61.1±0.3 |
| PGL | 66.2 | 68.8 | 0.0 | 0.0 | 68.8 | 70.2 | 33.8 | 45.6 | 82.8 | 85.9 | 5.3 | 10.0 | 72.0 | 73.9 | 24.5 | 36.8 | 74.1 | 76.1 | 25.0 | 35.2 |
| ROS | 52.3 | 51.5 | 73.0 | 60.4 | 65.6 | 65.3 | 72.2 | 68.6 | 54.1 | 53.6 | 65.5 | 58.9 | 60.3 | 59.8 | 71.6 | 65.2 | 62.0 | 61.6 | 72.4 | 66.2±0.3 |
| DANCE | 59.4 | 60.1 | 41.3 | 48.3 | 81.3 | 83.9 | 18.4 | 30.2 | 71.2 | 72.9 | 28.4 | 40.9 | 74.6 | 76.3 | 32.8 | 45.9 | 72.8 | 74.4 | 35.0 | 44.2±0.6 |
| DCC | - | - | - | 76.9 | - | - | - | 67.0 | - | - | - | 49.8 | - | - | - | 66.6 | - | - | - | 64.2 |
| LGU | 62.1 | 63.4 | 29.6 | 40.4 | 76.4 | 77.6 | 46.4 | 58.1 | 65.8 | 67.2 | 30.8 | 42.2 | 69.1 | 71.7 | 4.1 | 7.8 | 71.4 | 72.7 | 38.9 | 50.7 |
| OSLPP | 54.8 | 54.4 | 64.3 | 59.0 | 67.5 | 67.2 | 73.9 | 70.4 | 50.7 | 49.6 | 79.0 | 60.9 | 62.1 | 61.6 | 73.3 | 66.9 | 64.1 | 63.8 | 71.7 | 67.0 |
| **UADAL** | 52.0 | 51.1 | 74.5 | 60.6 | 69.4 | 69.1 | 78.3 | 73.4 | 54.5 | 53.5 | 80.5 | 64.2 | 62.8 | 62.1 | 78.8 | 69.5 | 63.1 | 62.6 | 78.0 | 68.7±0.2 |
| **cUADAL** | 52.5 | 51.8 | 71.1 | 59.9 | 69.5 | 69.3 | 76.3 | 72.6 | 54.9 | 53.8 | 82.0 | 65.0 | 61.8 | 61.1 | 77.4 | 68.3 | 63.1 | 62.5 | 77.6 | 68.5±0.1 |

Table 5: Classification accuracy (%) on Office-Home dataset using ResNet-50 as the backbone network. (bold: best performer, underline: second-best performer)