# OpenReview forum: "Unknown-Aware Domain Adversarial Learning for Open-Set Domain Adaptation"
_NeurIPS.cc/2022/Conference — NeurIPS 2022 Accept_

### Official Review · Reviewer_aB36 · 2022-07-05

**Rating:** 6
**Confidence:** 4
**Soundness:** 3 good
**Presentation:** 3 good
**Contribution:** 3 good

**Summary:**

This paper addresses open-set domain adaptation (ODA) problem, where target domain contains some unknown categories not seen in the source domain. Starting from the widely used adversarial training, the authors propose to introduce the known class recognition into it. Furthermore, the authors propose to deploy posterior inference to estimate the probability of belonging to unknown class. Thorough quantitative and analysis experiments prove the efficacy of the proposed method.

**Questions:**

1. Did the authors try using only one classifier head $C$?

As the first $\mathcal{C}_s$ dimension of $C$ can be directly utilized in the same way as $E$ did. Having only one unified classifier may improve the performance since two classifier heads may have disagreement.

2. Ablation study of entropy minimization in Eq. 19.

My feeling about this entropy minimization is that it may improve performance a lot. It will be interesting to see the effect of this term.

**Limitations:**

Maybe one limitation is that the posterior inference takes much time if with large dataset.

**Strengths And Weaknesses:**

## Strengths
1. Motivation of the paper is clear, that is, the unknown separation should be explicitly considered into the current adversarial training (more specifically STA [1]), which is the first contribution of the paper.
2. The probability of belonging to unknown is estimated by posterior inference, which is conducted on the whole target data. This avoid introducing extra hyperparameter to reject unknown categories as most existing ODA methods did. This is the second contribution.
3. The paper proposes to use $|\mathcal{C}_s| + 1$ -way classifier, which could directly lead to a decision boundary for unknown category. The deployed entropy minimization on it is expected to further improve the performance.
4. The experiments including both quantitative ones and other analysis, such as thresholding analysis, open-set recognition ability compared to other mechanism in Fig.6 and 7.


### Weakness

The method seems to be a little heavy, as the posterior inference demands access to all target data, and also the existing of two classifier heads ($E$ and $C$). Details in Questions below.


### Reference
[1]. Liu, Hong, et al. "Separate to adapt: Open set domain adaptation via progressive separation." Proceedings of the IEEE/CVF Conference on Computer Vision and Pattern Recognition. 2019.

---

> ### Author Response · Authors · 2022-08-02
> **Response to Reviewer aB36 - Part 2**
>
> **Q.** Maybe one limitation is that the posterior inference takes much time if with large dataset.
>
> **A.** In terms of complexity, the posterior inference increases the computational complexity because we need to fit the mixture model. As we already provided at 2.3.2 in Appendix, Wall-clock-time is increased as 5% with the posterior inference in the case of full data utilization. From this +5% increment, the performance has improved significantly than that without the posterior inference (as shown in Figure 7b in the paper). In addition, by utilizing the posterior inference, we avoid introducing any extra hyper-parameter to recognize the unknown instances, which is also our contribution.
>
> ***As an alternative, we can fit the mixture model only by sampling the target instances in order to reduce the computation time*** because the computational complexity is **$O(nk)$** where $n$ is the number of samples and $k$ is the number of fitting iterations (we use fixed $k$).
> **Figure 6 in Appendix 3.2.9 of the revised version** represents the Wall-clock time (s) and the performance measures, HOS (%), by varying the sampling ratio (%) for the target domain. Since the computational complexity is linearly increased by the number of samples ($n$), the wall-clock time is also linearly increased by increasing the sampling ratio. ***Interestingly, we observed that even though the sampling ratio is small, i.e. 10%, the performances of UADAL w.r.t. HOS does not decreased, on both Office-31 and Office-Home datasets.***
>
> In order to investigate the reason of this robustness on the sampling ratio, **we provide the qualitative analysis in Figure 7 in Appendix 3.2.9 of the revised version**. For each sampling ratio (50% and 10%), the left figure represents the original target entropy distribution, and the middle shows the sampled target entropy values and the fitted BMM densities. Finally, the right figure represents the weight distribution by the posterior inference.
> As you can see, ***even if the sampling ratio is small, the observation of the two modalities that the target-unknown instances have higher entropy values than the target-known instances still holds. Therefore, the mixture model is appropriately fitted even with the sampled target entropy values.*** From the right figures, we observed that the open-set recognition on the target domain, $\hat{w}$, is still informative, and it leads to maintain the performance of UADAL in the small sampling ratio.
>
> **We sincerely request the reviewer aB36 to refer the Figure 6 and Figure 7 in Appendix 3.2.9 of the revised version.**

---

> > ### Comment · Reviewer_aB36 · 2022-08-03
> > **Thanks for the reply**
> >
> > The authors address all my questions, and thanks a lot for their additional experiments and clarification. I will keep my score as weak accept 6.

---

> > > ### Author Response · Authors · 2022-08-05
> > > **Thanks to the reviewer aB36.**
> > >
> > > Again, thanks for your work on the reviews and the feedback. Please leave a comment if you have further questions.

---

> ### Author Response · Authors · 2022-08-02
> **Response to Reviewer aB36 - Part 1**
>
> **Q.** [Question 1] Using only one classifier head $C$ as the first $C_s$ dimension of $C$ can be directly utilized in the same way as $E$ did.
>
> **A.** Using the classifier $C$ (except the last unknown dimension) directly is also feasible, as $E$ does. Although it is feasible, however, it leads to wrong decisions for the target-unknown instances. As we explained in the paper, the network $E$ learns the decision boundary over $C_s$ classes while the network $C$ does over $C_{s}+1$ including *unknown* class. Especially, in the case of the target-unknown instances (**higher $\hat{w}$**), **the network $C$ is enforced to classify the instances as *unknown* class, which is $C_{s}+1$-th dimension.** When optimizing the network $C$ with the target-unknown instances, we expect that the output of the network $C$ would have a higher value on the unknown dimension. With this point, if we use the first $C_s$ dimension of $C$ to calculate the entropy values for the target-unknown instances, **there is no evidence that the distribution of the $C$'s output is flat over ${C}_s$ classes** where the flat output implies to a higher entropy value. Even though the largest predicted value except the last dimension is small, the entropy value might be lower due to imbalance in the output over the first $C_s$ dimension. Then, it becomes to be considered as \textit{known} class, which is wrong decision for the target-unknown instances. Therefore, it gives the negative effects on the open-set recognition, and it adversely affects when training the model.
>
> **We conduct the experiments to compare using $E$ or $C$ for the entropy, as an ablation study.** The experimental results on both cases are shown in **Table 1 of Appendix 3.2.7 of the revised version**, applied to Office-31 and Office-Home datasets. The network $E$ in Table 1 is the current UADAL model and $C$ represents that the entropy values are generated by the classifier $C$ without introducing the network $E$. As you can see, **the performance when utilizing $E$ (original UADAL) is better than when replacing $E$ by $C$**. It means that the network $E$ learns the decision boundary for the known classes, and it leads to recognize the open-set instances effectively. It should be noted that we utilize the structure of $E$ as an one-layered network to reduce the computation burden.
>
> (We provide Table 1 of Appendix 3.2.7 of the revised version also at the below comment, for convenience.)
>
> **Q.** [Question 2] Ablation study of entropy minimization in Eq. 19.
>
> **A.** . The entropy minimization is important part for the fields such as semi-supervised learning [3, 4] and domain adaptation [5,6,7,8] where the label information of the dataset is not available. In order to show the effect of this term, **we conduct the ablation study on the Entropy Minimization for the Office-31 and Office-Home**. We provide the experimental results in **Table 2 of Appendix 3.2.8** of the revised version. Combined with the results in the main paper, ***the experimental results show that UADAL without the entropy minimization loss still performs better than other baselines.*** It represents that UADAL learns the feature space appropriately as we intended to suit Open-Set Domain Adaptation. The properly learned feature space leads to effectively classify the target instances without the entropy minimization.
>
> (We provide Table 2 of Appendix 3.2.8 of the revised version also at the below comment, for convenience.)
>
> [3] Grandvalet, Yves, and Yoshua Bengio. "Semi-supervised learning by entropy minimization." Advances in neural information processing systems 17 (2004).
>
> [4] Sohn, Kihyuk, et al. "Fixmatch: Simplifying semi-supervised learning with consistency and confidence." Advances in neural information processing systems 33 (2020): 596-608.
>
> [5] Long, Mingsheng, et al. "Unsupervised domain adaptation with residual transfer networks." Advances in neural information processing systems 29 (2016).
>
> [6] Liu, Hong, et al. "Separate to adapt: Open set domain adaptation via progressive separation." Proceedings of the IEEE/CVF Conference on Computer Vision and Pattern Recognition. 2019.
>
> [7] Saito, Kuniaki, et al. "Universal domain adaptation through self supervision." Advances in neural information processing systems 33 (2020): 16282-16292.
>
> [8] Bucci, Silvia, Mohammad Reza Loghmani, and Tatiana Tommasi. "On the effectiveness of image rotation for open set domain adaptation." European Conference on Computer Vision. Springer, Cham, 2020.

---

> > ### Author Response · Authors · 2022-08-02
> > **Table 2 of Appendix 3.2.8 of the revised version**
> >
> > We provide Table 2 in Appendix 3.2.8 at this comment, for convenience.
> > ## Table 2 in Appendix 3.2.8
> > ### 1) Office-31 (ResNet-50)
> > | |      | A-W  |      |      | A-D  |      |      | D-W  |      |      | W-D   |      |      | D-A  |      |      | W-A  |      |      | AVG  |      |
> > |:------------------------:|------|------|------|------|------|------|------|------|------|------|-------|------|------|------|------|------|------|------|------|------|------|
> > | **Entropy**                | OS*  | UNK  | HOS  | OS*  | UNK  | HOS  | OS*  | UNK  | HOS  | OS*  | UNK   | HOS  | OS*  | UNK  | HOS  | OS*  | UNK  | HOS  | OS*  | UNK  | **HOS**  |
> > |  **X** | 85.9 | 84.4 | 85.1 | 84.7 | 83.6 | 84.2 | 95.6 | 98.9 | 97.2 | 98.7 | 100.0 | 99.3 | 75.2 | 86.0 | **80.3** | 72.6 | 87.4 | **79.3** | 85.4 | 90.0 | 87.5 |
> > |  **O** | 84.3 | 94.5 | **89.1** | 85.1 | 87.0 | **86.0** | 99.3 | 96.3 | **97.8** | 99.5 | 99.4  | **99.5** | 73.3 | 87.3 | 79.7 | 67.4 | 88.4 | 76.5 | 84.8 | 92.1 | **88.1** |
> >
> > ### 2) Office-Home (ResNet-50)
> >
> > |   |      | P-R  |      |      | P-C  |      |      | P-A  |      |      | A-P  |      |      | A-R  |      |      | A-C  |      |
> > |:---:|------|------|------|------|------|------|------|------|------|------|------|------|------|------|------|------|------|------|
> > |  **Entropy** | OS*  | UNK  | HOS  | OS*  | UNK  | HOS  | OS*  | UNK  | HOS  | OS*  | UNK  | HOS  | OS*  | UNK  | HOS  | OS*  | UNK  | HOS  |
> > |  **X** | 70.7 | 78.2 | 74.2 | 48.4 | 76.6 | **59.3** | 49.9 | 76.2 | 60.3 | 64.5 | 79.4 | **71.2** | 78.8 | 75.2 | 77.0 | 56.3 | 75.1 | **64.3** |      |      |      |   |
> > |  **O** | 71.6 | 83.1 | **76.9** | 43.4 | 81.5 | 56.6 | 50.5 | 83.7 | **63.0** | 69.1 | 72.5 | 70.8 | 81.3 | 73.7 | **77.4** | 54.9 | 74.7 | 63.2 |      |      |      |   |
> >
> > |   |      | R-A  |      |      | R-P  |      |      | R-C  |      |      | C-R  |      |      | C-A  |      |      | C-P  |      |      | AVG  |      |
> > |:---:|------|------|------|------|------|------|------|------|------|------|------|------|------|------|------|------|------|------|------|------|------|
> > | **Entropy**  | OS*  | UNK  | HOS  | OS*  | UNK  | HOS  | OS*  | UNK  | HOS  | OS*  | UNK  | HOS  | OS*  | UNK  | HOS  | OS*  | UNK  | HOS  | OS*  | UNK  | **HOS**  |
> > |  **X**  | 62.3 | 76.4 | 68.6 | 71.3 | 81.0 | 75.8 | 57.3 | 65.6 | **61.2** | 68.1 | 75.9 | 71.8 | 54.7 | 70.9 | 61.7 | 60.1 | 72.6 | 65.8 | 61.9 | 75.2 | 67.6 |
> > |  **O**   | 66.7 | 78.6 | **72.1** | 77.4 | 76.2 | **76.8** | 51.1 | 74.5 | 60.6 | 69.1 | 78.3 | **73.4** | 53.5 | 80.5 | **64.2** | 62.1 | 78.8 | **69.5** | 62.6 | 78.0 | **68.7** |

---

> > ### Author Response · Authors · 2022-08-02
> > **Table 1 of Appendix 3.2.7 of the revised version**
> >
> > We provide Table 1 of Appendix 3.2.7 of the revised version at this comment, for convenience.
> > ## Table 1 of Appendix 3.2.7
> > ### 1) Office-31 (ResNet-50)
> > | |      | A-W  |      |      | A-D  |      |      | D-W  |      |      | W-D   |      |      | D-A  |      |      | W-A  |      |      | AVG  |      |
> > |:------------------------:|------|------|------|------|------|------|------|------|------|------|-------|------|------|------|------|------|------|------|------|------|------|
> > |     **Network**     | OS*  | UNK  | HOS  | OS*  | UNK  | HOS  | OS*  | UNK  | HOS  | OS*  | UNK   | HOS  | OS*  | UNK  | HOS  | OS*  | UNK  | HOS  | OS*  | UNK  | **HOS**  |
> > | **C**                  | 80.5 | 92.4 | 86.1 | 79.4 | 91.0 | 84.8 | 99.0 | 98.3 | **98.6** | 98.7 | 100.0 | 99.3 | 68.7 | 89.7 | 77.9 | 56.5 | 89.1 | 68.9 | 80.5 | 93.4 | 85.9 |
> > | **E**                     | 84.3 | 94.5 | **89.1** | 85.1 | 87.0 | **86.0** | 99.3 | 96.3 | 97.8 | 99.5 | 99.4  | **99.5** | 73.3 | 87.3 | **79.7** | 67.4 | 88.4 | **76.5** | 84.8 | 92.1 | **88.1** |
> >
> > ### 2) Office-Home (ResNet-50)
> >
> > |  |      | P-R  |      |      | P-C  |      |      | P-A  |      |      | A-P  |      |      | A-R  |      |      | A-C  |      |
> > |:---:|------|------|------|------|------|------|------|------|------|------|------|------|------|------|------|------|------|------|
> > | **Network**   | OS*  | UNK  | HOS  | OS*  | UNK  | HOS  | OS*  | UNK  | HOS  | OS*  | UNK  | HOS  | OS*  | UNK  | HOS  | OS*  | UNK  | HOS  |
> > |  **C** | 68.8 | 83.7 | 75.5 | 39.3 | 79.4 | 52.6 | 47.9 | 84.3 | 61.1 | 65.0 | 76.3 | 70.2 | 78.4 | 75.6 | 77.0 | 46.7 | 77.2 | 58.2 |
> > |  **E** | 71.6 | 83.1 | **76.9** | 43.4 | 81.5 | **56.6** | 50.5 | 83.7 | **63.0** | 69.1 | 72.5 | **70.8** | 81.3 | 73.7 | **77.4** | 54.9 | 74.7 | **63.2**|
> >
> > |   |      |  R-A    |   | |   R-P   |      |   |   R-C   |      |   |    C-R  |      |   |   C-A   |      |   |   C-P   |      |   |   AVG   |      |
> > |:------:|------|------|------|------|------|------|------|------|------|------|------|------|------|------|------|------|------|------|------|------|------|
> > | **Network** | OS*  | UNK  | HOS  | OS*  | UNK  | HOS  | OS*  | UNK  | HOS  | OS*  | UNK  | HOS  | OS*  | UNK  | HOS  | OS*  | UNK  | HOS  | OS*  | UNK  | **HOS**  |
> > |  **C** | 64.2 | 79.2 | 71.0 | 75.5 | 78.6 | **77.0** | 44.9 | 71.7 | 55.2 | 63.0 | 75.0 | 68.5 | 49.9 | 77.4 | 60.7 | 57.8 | 79.2 | 66.8 | 58.4 | 78.1 | 66.1 |
> > |  **E** | 66.7 | 78.6 | **72.1** | 77.4 | 76.2 | 76.8 | 51.1 | 74.5 | **60.6** | 69.1 | 78.3 | **73.4** | 53.5 | 80.5 | **64.2** | 62.1 | 78.8 | **69.5** | 62.6 | 78.0 | **68.7** |

---

### Official Review · Reviewer_whZG · 2022-07-11

**Rating:** 5
**Confidence:** 4
**Soundness:** 2 fair
**Presentation:** 2 fair
**Contribution:** 3 good

**Summary:**

This paper proposed an unknown-aware domain adversarial learning framework for open-set domain adaptation (OSDA), where the target domain has a larger label set than that of the source domain. This also means that the model needs to see novel class samples during the adaptation, which brings extra difficulty for domain adversarial learning. This work tackles this problem by introducing a three-dim discriminator for aligning the source (Ds) and target known (Tk) while separating the target unknown (Tu). The authors provide both theoretical and experimental analysis to support this type of domain adversarial learning and their methods are compared with state-of-the-art OSDA works.

**Questions:**

1.Regarding weakness 1, how can UADAL handle situations where: (1) Entropy(Tu) ≈ Entropy(Tk), under the case that there is a clear domain shift between the source and target domains; (2) Entropy(Tu) < Entropy(Tk), where there might exist target-unknown samples that yield lower entropy than the target-known ones and cause possibly negative effect when models are optimized via Eq. (11) and Eq. (12).

2.It is better to provide not just HOS but also OS, OS* and UNK for the Office-31 and Office-Home benchmarks (i.e., Table 1), as some recent and previous methods did not include HOS as the metric when reporting results.

3.Comparisons with some recent OSDA works are missing, including JPOT (IJCAI’20) on Office-31 and Office-Home and LGU (WACV’22) on VisDA.

References:

JPOT: Xu, et al. “Joint Partial Optimal Transport for Open Set Domain Adaptation,” IJCAI, 2020.

LGU: Baktashmotlagh, et al. “Learning to Generate the Unknowns as a Remedy to the Open-Set Domain Shift,” WACV, 2022.


###################### After rebuttal ###################

I would like to thank the authors' detailed comments which well addressed my concerns. I will keep my score unchanged.



**Ethics Review Area:**

["I don’t know"]

**Limitations:**

The limitations stated in the supplementary materials, i.e., suppL281 – 289, are the limitations for the general DA problem, but not specifically for their own methods. It is the current paradigm of DA that cannot tackle streamlined data gathering and needs sufficient source domain categories, not just for this paper. More concrete limitations of the proposed methods in this paper are recommended included.

**Strengths And Weaknesses:**

Strengths:

1.This work proposed an unknown-aware domain adversarial learning (UADAL) framework to handle the OSDA problem and the idea of aligning Ds and Tk while separating Tu via explicit adversarial learning is new.

2.This work provides theoretic analysis of the optimization process for the proposed UADAL.

3.The experimental results show that the proposed UADAL can achieve better performance than the state-of-the-art OSDA methods.

Weaknesses:

1.The framework is based on the assumption that (as in L192) Tu will have higher entropy than Tk, which might be too strong to hold.

2.Domain adversarial learning is known to be unstable during optimization and lacks interpretability.

---

> ### Author Response · Authors · 2022-08-02
> **Response to Reviewer whZG - Part 2**
>
> **Q.** [ Question 2] It is better to provide not just HOS but also OS, OS* and UNK for the Office-31 and Office-Home benchmarks (i.e., Table 1), as some recent and previous methods did not include HOS as the metric when reporting results.
>
> **A.** As a reminder, HOS metric is a harmonic mean of OS* and UNK where OS* is accuracy for the known class classification and UNK is for the unknown classification. Since Open-Set Domain Adaptation should perform well on both tasks, we choose HOS metric as a primary metric. For other metrics such as OS, OS*, and UNK, we omit these due to space limit in the paper.
>
> In this rebuttal period, **we provide the full experimental results including OS, OS$^{*}$, and UNK in Appendix 3.2.10 of the revised version**.
>
> **Q.** [ Question 3] Comparisons with some recent OSDA works are missing, including JPOT (IJCAI’20) on Office-31 and Office-Home and LGU (WACV’22) on VisDA.
>
> **A.** We added the comparisons with the recent work, LGU, on Office-Home and VisDA by utilizing their reported performances (Please refer to **Table 3 in Appendix 3.2.10**). It should be noted that ***we still have the best performances*** even we included these additional baseline, LGU.
>
> (We provide Table 3 in Appendix 3.2.10 of the revised version at the below link, for convenience. https://openreview.net/forum?id=IwC_x50fvU&noteId=eRqr5GXqUsS)
>
> In terms of JPOT, we do not include JPOT as a baseline. JPOT provides OS and OS* measures in the original paper. Given these two metrics, we need to calculate UNK and then HOS metrics since our primary metric is HOS. Then, UNK is calculated as follows, $UNK = (C_s+1) \times OS - C_s \times OS^{\*}$ with the number of known classes ($C_s$). With this formula, ***we found that there are some cases where UNK (unknown classification accuracy) is higher than 100%, which is infeasible.*** For example, the OS and OS* on D→W of Office-31 is 98.1 and 96.2, respectively. Then, UNK is calculated as UNK=11$\times$98.1-10$\times$96.2 = 117.1 (%). With this reason, we judged that UNK and HOS metrics from JPOT are not reliable to compare with UADAL. ***Therefore, we do not include JPOT in our comparison.***
>
> **Q.** [ Limitation] More concrete limitations of the proposed methods in this paper are recommended included.
>
> **A.** Our work solves the Open-Set Domain Adaptation problem. It intrinsically assumes the existence of ‘unknown’ information in the target domain. Our open-set recognition is based on this assumption, thus we fit the mixture model where each mode represents for known/unknown information. We think that the common assumption of the high entropy value on target-unknowns could be considered as a limitation, as well. **We modified the limitations at the Section 4 in Appendix of the revised version.**

---

> ### Author Response · Authors · 2022-08-02
> **Response to Reviewer whZG - Part 1**
>
> **Q.** [ Weakness 1] The framework is based on the assumption that (as in L192) $tu$ will have higher entropy than $tk$, which might be too strong to hold.
>
> **A.** In Open-Set Domain Adaptation, there are disjoint class splits between target-known and target-unknown classes. Even though there is a domain shift between the source and the target domain, the information in the target-known classes is shared with the classes in the source domain. On the other hand, the target-unknown instances do not share common features with known classes. From this perspective, it is natural to assume that the entropy values from the decision boundary by the known classes are higher on target-unknown ($tu$) instances than target-known ($tk$) instances. In literature, ***this assumption has been common in the Open-Set Domain Adaptation [1,2]***. Given the classifier to learn the decision boundary over the known classes, UAN [2] hypothesis that the target-unknown samples have higher entropy values than the target-known samples, and they showed that the hypothesis is successfully justified. DANCE [1] also utilizes the entropy value of a classifier’s output to draw a boundary between “known” and “unknown” points.
>
> ***Certainly, we verified this assumption empirically in Figure 6 of the paper which clearly shows the two modalities of the entropy value distribution, corresponding to the target-known (left mode) and the target-unknown (right mode) as our assumption.*** Specifically, a mode of target-known has the average entropy of **0.29 (std=0.12)** whereas the other mode of target-unknown has the average entropy of **0.80 (std=0.12)**, so the difference is clear. This statistics are obtained from the initial set-up of the network $E$ (by Eq. 20 in the submitted paper) on D->W task in Office-31 dataset (the case in Figure 6 of the paper). ***From this empirical observation, we found that our assumption empirically holds.*** Therefore, we modeled the posterior inference from the mixture of two Beta distributions for open-set recognition.
> *We also observed that this assumption still holds even if we sampled the target instances such as 50% and 10%, as shown at **Figure 7 in Appendix 3.2.9 of the revised version** (The details are in the following response: https://openreview.net/forum?id=IwC_x50fvU&noteId=l1tJ6X21HxL ).
>
> [1] Saito, Kuniaki, et al. "Universal domain adaptation through self supervision." Advances in neural information processing systems 33 (2020): 16282-16292.
>
> [2] You, Kaichao, et al. "Universal domain adaptation." Proceedings of the IEEE/CVF conference on computer vision and pattern recognition. 2019.
>
> **Q.** [ Weakness 2] Domain adversarial learning is known to be unstable during optimization and lacks interpretability.
>
> **A.** **Figure 8** in the paper shows the empirical convergence of UADAL during the optimization procedure. **Theorem 3.1** claims that the proposed unknown-aware domain adversarial learning leads to obtain the domain-invariant features over the source and the target-known domain, while segregating the target-unknown features, which is formalized as $p_s$$≈$$p_{tk}$ and $p_{tu}$$↔${$p_s$,$p_{tk}$}. ***Figure 8 confirms this theoretic claim*** by showing 1) the closeness between $s$ and $tk$, i.e., $p_s$$≈$$p_{tk}$, and 2) the distantness between $s$ and $tu$, i.e., $p_{s}$$↔$$p_{tu}$. ***Given this alignment between the empirical result and the theoretic claim, we concluded that the proposed adversarial learning was able to achieve the intended convergence.***
>
> **Q.** [ Question 1] how can UADAL handle situations where: (1) Entropy(Tu) ≈ Entropy(Tk), (2) Entropy(Tu) < Entropy(Tk).
>
> **A.** First of all, we ask the reviewer to examine **the entropy distribution of $tk$ and $tu$ (including $s$) in Figure 6 in the paper** because the reviewer is asking us to assume contradictory situation to our empirical statistics in Figure 6 in the paper.
>
> (1) Having said that, if we are being forced to assume that the entropy values of $tk$ and $tu$ are similar, the posterior inference of UADAL will automatically detect and adjust the threshold between the entropy values of $tk$ and $tu$. Therefore, ***if the entropy values of $tu$ are still higher than $tk$ with slight margin, UADAL will be able to function.*** This is the main difference between UADAL and existing baselines relying on fixed thresholds.
>
> (2) If we have to assume that the entropy values of $tu$ is lower than $tk$, UADAL will not able to function because these phenomena indicate that the classifier $E$ has clearer decision boundary on the $tu$ than $tk$, and this directly contradicts to our assumptions in Line 192, our main paper. However, this is ***an unrealistic scenario*** because “target-known” are known classes with well-defined decision boundary in the open-set recognition settings (Please refer the first answer in this response).

---

> ### Author Response · Authors · 2022-08-08
> **Dear Reviewer whZG**
>
> Again, thanks for your work on the reviews and the feedback. Please leave a comment if you have further questions. We will be happy to provide additional clarification. Thank you so much for your time.

---

### Official Review · Reviewer_31hV · 2022-07-11

**Rating:** 5
**Confidence:** 4
**Soundness:** 3 good
**Presentation:** 3 good
**Contribution:** 3 good

**Summary:**

This paper proposes to deal with the open-set domain adaptation problem by not only considering the known classes matching but also the target-unknown distribution segregating. Both the theoretical analysis and corresponding algorithm for dealing with the problem are provided. in general, the paper is well-motivated, and the experiments are solid.

**Questions:**

+ what are the key differences in the insights between the proposed method and previous methods?
+ Why do some baseline methods only achieve very low performance? the reliability of the experimental comparisons are questionable.


###################### After rebuttal ###################

The authors have addressed my main concerns. I will keep my original score as borderline accept.

**Strengths And Weaknesses:**

Strengths
+ The proposed method has some novel ideas that separate the target unknown classes from the shared classes in the feature space.
+ The estimation of the open-set recognition is achieved by posterior inference, which is interesting.
+ Theoretical analysis is provided.
+ The experiments are extensive and good results are achieved on several benchmark datasets.

Weaknesses

- The paper a little bit over claim their main observation and insights. The baseline methods, such as STA also claimed the unknown classes should be separated from the known ones. Though the proposed method is very different from STA as compared and discussed by the authors, the key insights are not brand new.

- Though the difference between the proposed method and STA is compared, it is still unclear why the proposed method by separating the unknown classes from the known ones in the feature space with an adversarial learning framework is superior to STA with a classifier. More insights, explanations, and analyses would be better.

- In the experiments, the implementation of other baseline methods is questionable. For example, the DANCE method only achieves very low performance on the Office-Home dataset with the EfficientNet backbone. It is unclear why the very low implementation results are achieved, leading to unreliable comparisons.

---

> ### Author Response · Authors · 2022-08-02
> **Response to Reviewer 31hV - Part 2**
>
> **Q.** [ Weakness 3] In the experiments, the implementation of other baseline methods is questionable. It is unclear why the very low implementation results are achieved, leading to unreliable comparisons.
>
> **A.** For reliable and fair comparisons, we provide the summary table of the experimental results with the officially reported performances of the baselines, which is denoted as *. ***Table 3 in Appendix 3.2.10 of the revised version shows that UADAL outperforms the baselines over all datasets, in the conventional setting of the backbone networks (such as Office-31/Office-Home with ResNet-50 and VisDA with VGGNet)***.
>
> In order to show the robustness of the architecture choices, as we mentioned in the paper, we expanded the choices of the pre-trained backbone networks to EfficientNet and DenseNet. These backbone networks have not been used in OSDA problem. **Therefore, there are no known hyperparameter settings for the baselines with EfficientNet and DenseNet.**
>
> The below is the detailed answer on the lower performances of baselines, especially DANCE with EfficientNet. As we said, there is no reported performance of DANCE with the additional backbone choices. Therefore, we implemented additional variants of DANCE with EfficientNet and DenseNet by following their officially released codes. **In order to compare fairly, we set all hyper-parameters with that of ResNet-50 case as UADAL is being set.** Specifically, DANCE requires a threshold value (\rho) to decide whether a target instance belongs to “known” class or not, which is very sensitive to the performance. We confirmed that they utilize the different values over the experimental settings. **This hyperparameter sensitivity may degrade the performance of DANCE.** Unlike DANCE, ***UADAL does not require a threshold setting because it has a posterior inference to automatically find the threshold to decide open-set instances. Therefore, this becomes the key reason behind the performance difference.***
>
> **Once again, please refer to the summarized table for the reliable experimental comparisons, in Table 3 of Appendix 3.2.10 of the revised version.**
>
> (We provide Table 3 in Appendix 3.2.10 of the revised version also at this below comment, for convenience.)

---

> > ### Author Response · Authors · 2022-08-02
> > **Table 3 in Appendix 3.2.10 of the revised version**
> >
> > We provide Table 3 in Appendix 3.2.10 of the revised version at this comment, for convenience.
> >
> > ## Table 3 in Appendix 3.2.10
> > ### Summary of the experimental results.
> > |        |      |   Office-31   |   (Res50)  |       |      |    Home   |  (Res50)     |   |   |    VisDA  |   (VGG)    |      |
> > |:--------:|:--------:|:--------:|:--------:|:--------:|:--------:|:--------:|:--------:|:--------:|:--------:|:--------:|:--------:|:--------:|
> > |        | OS   | OS*  | UNK  | **HOS**  | OS   | OS*  | UNK  | **HOS**  | OS   | OS*  | UNK  | **HOS**  |
> > | DANN   | 85.4 | 87.1 | 68.3 | 75.9 | 53.5 | 52.6 | 77.1 | 60.7 | -    | -    | -    | -    |
> > | CDAN   | 86.1 | 88.3 | 63.9 | 73.4 | 55.3 | 54.5 | 74.6 | 61.4 | -    | -    | -    | -    |
> > | OSBP*   | 86.6 | 87.2 | 80.4 | 83.7 | 64.2 | 64.1 | 66.3 | 64.7 | 62.9 | 59.2 | 85.1 | 69.8 |
> > | STA*    | 82.5 | 84.3 | 64.8 | 72.5 | 61.9 | 61.8 | 63.3 | 61.1 | 66.8 | 63.9 | 84.2 | 72.7 |
> > | PGL*   | 81.1 | 82.7 | 64.7 | 72.6 | 74.1 | 76.1 | 25.0 | 35.2 | 80.7 | 82.8 | 68.1 | 74.7 |
> > | ROS*    | 86.5 | 86.6 | 85.8 | 85.9 | 62.0 | 61.6 | 72.4 | 66.2 | -    | -    | -    | -    |
> > | DANCE  | 91.0 | 94.0 | 60.2 | 73.1 | 72.8 | 74.4 | 35.0 | 44.2 | -    | -    | -    | -    |
> > | DCC*    | -    | -    | -    | 86.8 | -    | -    | -    | 64.2 | 68.8 | 68.0 | 73.6 | 70.7 |
> > | LGU*    | -    | -    | -    | -    | 71.4 | 72.7 | 38.9 | 50.7 | 70.1 | 69.2 | 75.5 | 72.2 |
> > | OSLPP*  | 89.0 | 89.3 | 85.6 | 87.4 | 64.1 | 63.8 | 71.7 | 67.0 | -    | -    | -    | -    |
> > | **UADAL**  | 85.5 | 84.8 | 92.1 | 88.1 | 63.1 | 62.6 | 78.0 | **68.7** | 67.4 | 63.1 | 93.3 | 75.3 |
> > | **cUADAL** | 85.6 | 84.8 | 93.0 | **88.5** | 63.1 | 62.5 | 77.6 | 68.5 | 68.3 | 64.3 | 92.6 | **75.9** |
> >
> > * The results in Office-31 and Office-Home are the averaged accuracies over the tasks (with three times repetitions) because there are the multiple domains.
> > * The detailed results over the each tasks in Office-31 and Office-Home are provided at Table 4 and 5 in  Appendix 3.2.10 of the revised version.
> > * $*$ stands for the officially reported performances.

---

> ### Author Response · Authors · 2022-08-02
> **Response to Reviewer 31hV - Part 1**
>
> **First of all, we sincerely request the reviewer 31hV to refer the Figure 5 in Appendix 3.2.6 of the revised version**, in terms of our responses to your questions.
>
> **Q.** [ Weakness 1 ] The paper a little bit over claim their main observation and insights. Though the proposed method is very different from STA as compared and discussed by the authors, the key insights are not brand new.
>
> **A.** Open-Set Domain Adaptation essentially requires segregating the target-unknown ($tu$) classes from the target-known ($tk$) ones. As a reviewer 31Hv said, STA also proposed to separate the target-unknown instances. However, this implicit separation does not theoretically guarantee the optimal states of separation in the feature distributions. We claimed that the distribution segregation of $tu$ has been overestimated in previous works. ***Our key insight is that the existing methods are not suitable to segregate $tu$ for OSDA.*** ***Therefore, our work is the first to explicitly segregate the target-unknown distribution while aligning the source and the target-known distribution in a unified domain adversarial learning framework.*** **This is our novel contribution, which is consensus with the opinion by the reviewer whZG and aB36 (in each strength 1).** This explicit segregation is qualitatively observed in the t-SNE view of Figure 3 and the distribution distance metric (PAD) of Figure 4 in the paper, and we achieved significantly better performances than STA as reported in Table 1 and 2 in the paper. ***The performance gain is notable in the UNK metric (see Table 2 in the paper or Table 3 in the appendix of the revised version), which gets direct benefit from the explicit segregation from UADAL, which it is not feasible by STA.*** We also provide the theoretical guarantee to the unknown-aware feature alignments for OSDA, which is unexplored in OSDA field.
>
> **Q.** [ Weakness 2 ] It is still unclear why the proposed method by separating the unknown classes from the known ones in the feature space with an adversarial learning framework is superior to STA with a classifier. More insights, explanations, and analyses would be better.
>
> **A.** In order to provide more insights and explanations compared to STA, we provide **the additional analysis in Figure 5 in Appendix 3.2.6 of the revised version**. Key difference to STA is that UADAL explicitly segregates the target-unknown feature distribution, while STA conducts implicit separation. **Therefore, we claimed that this explicit segregation of UADAL is the essential part for OSDA, and leads better performance.** In order to investigate this effect, ***we provide an analysis on the correlation between the evaluation metric and the distance measure of the feature distribution.*** We utilize **UNK, and HOS** as evaluation metrics, and **PAD between Tk and Tu** as the distance measure. **Figure 5 in Appendix 3.2.6** shows the scatter plots of (UNK & PAD) and (HOS & PAD). The grey arrows in the figure mean the corresponding tasks by UADAL and STA. As shown in the figure, **HOS and UNK has a positive correlation with the distance between the target-unknown ($tu$) and the target-known ($tk$)**. ***In simple words, better segregation enables better HOS and UNK***. Specifically, from STA to UADAL on the same task, the distances (PAD on $tk$/$tu$) are increased, and the corresponding performances, UNK and HOS, are also increased. It means that UADAL effectively segregates the target feature distributions, and then leads to better performance for OSDA. In other words, the proposed unknown-aware feature alignment is important to solve OSDA problem. ***From this analysis, our explicit segregation of tu is our main contribution, and it is key difference to the previous methods.***
>
> We hope that our responses resolve your questions.

---

### Author Response · Authors · 2022-08-02
**Overall Comments**

First of all, we sincerely thank you for your work on the reviews and the feedback. We clarify that we revise and update Appendix, named as **appendix_revised.pdf** in the Supplementary Material. The revised version includes the added tables, figures, and texts on our responses. Therefore, we ask the reviewers to read the parts in the revised version of appendix, mentioned in our responses.

---

### Meta-Review · Area_Chair_nxJh · 2022-08-26

**Recommendation:** Accept
**Confidence:** Certain

**Metareview:**

This paper proposes a novel method called UADAL (Unknown-Aware Domain Adversarial Learning) for Open-Set Domain Adaptation (OSDA). Specifically, the proposed method performs source and target-known distribution alignment while simultaneously separating source and target-unknown distributions in the feature alignment procedure. In the OSDA, the idea of source and target-known distribution alignment while simultaneously separating the source and target-unknown distributions through explicit adversarial learning is novel. Furthermore, a theoretical analysis of the optimization process of the proposed UADAL is conducted. All three reviewers had similar positive comments on this paper and were satisfied with the authors' feedback. Thus the AC recommends it for acceptance.

**Award:**

No

---

### Decision · Program_Chairs · 2022-09-14

Accept